# Causal Discovery over Clusters of Variables in Markovian Systems

**Tara V Anand**
Department of Biomedical Informatics
Columbia University
tara.v.anand@columbia.edu

**Adèle H Ribeiro**
Institute of Medical Informatics
University of Münster
adele.ribeiro@uni-muenster.de

**Jin Tian**
Mohamed bin Zayed University
of Artificial Intelligence
jin.tian@mbzuai.ac.ae

**George Hripcsak**
Department of Biomedical Informatics
Columbia University
gh13@cumc.columbia.edu

**Elias Bareinboim**
Causal Artificial Intelligence Laboratory
Columbia University
eb@cs.columbia.edu

## Abstract

Causal discovery methods are powerful tools for uncovering the structure of relationships among variables, yet they face significant challenges in scalability and interpretability, especially in high-dimensional settings. In many domains, researchers are not only interested in causal links between individual variables, but also in relationships among sets or clusters of variables. Learning causal structure at the cluster level can both reveal higher-order relationships of interest and improve scalability. In this work, we introduce an approach for causal discovery over clusters in Markov causal systems. We propose a new graphical model that encodes knowledge of relationships between user-defined clusters while fully representing independencies and dependencies over clusters, faithful to a given distribution. We then define and characterize a graphical equivalence class of these models that share cluster-level independence information. Lastly, we present a sound and complete algorithm for causal discovery to represent learnable causal relationships between clusters of variables.

## 1 Introduction

Causal discovery, where observational data are used to uncover causal relationships between variables, is a task of interest in many domains [13, 19]. The goal in causal discovery is to use data to learn as much information as possible about the underlying causal diagram, a graph that illustrates assumptions about the presence and direction of causal and confounding relationships between variables in a system. One approach to causal discovery has been through constraint-based methods, where independence information, combined with logic regarding graphical properties, are used to determine structural properties of the graph, and constraints on possible causal diagrams that could correspond with the dataset [22, 13, 18]. Among constraint-based algorithms, PC is a foundational algorithm for Markovian systems, where causal sufficiency, or the absence of latent confounding, is assumed [19] and there are several extensions of this algorithm [15, 17]. FCI is the comparable algorithm for non-Markovian settings where unobserved confounding is permitted [26, 20] and of which there are also

39th Conference on Neural Information Processing Systems (NeurIPS 2025).

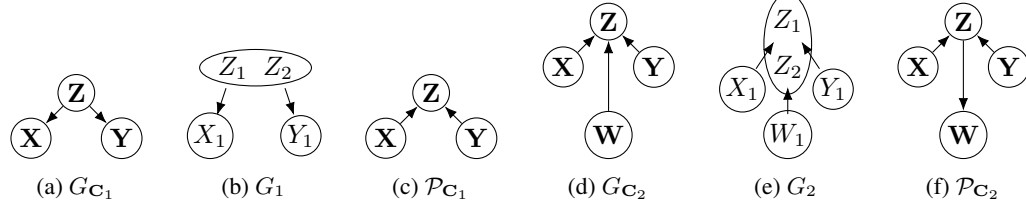

(a) $G_{\mathbf{C}_1}$     (b) $G_1$     (c) $\mathcal{P}_{\mathbf{C}_1}$     (d) $G_{\mathbf{C}_2}$     (e) $G_2$     (f) $\mathcal{P}_{\mathbf{C}_2}$

Figure 1: DAGs $(b)$ and $(e)$ in the classes represented by C-DAGs $(a)$ and $(d)$, respectively. $(c)$: an attempted graphical equivalence class for $(a)$ after applying a collider search test given a distribution from $G_1$. $(f)$: an attempted graphical equivalence class for $G_{\mathbf{C}_2}$ after applying a modified collider search test requiring $\mathbf{X} \not\perp\!\!\!\perp \mathbf{Y}|\mathbf{Z}$, and applying an orientation rule, given a distribution from $G_2$.

several extensions [7, 14]. Typically, the data constraints are insufficient for uniquely identifying a causal diagram. Instead, the graphical object of interest, and the output of causal discovery algorithms, is an equivalence class of causal diagrams that fully encodes the data constraints.

In both Markovian and non-Markovian systems, however, existing algorithms are often computationally prohibitive with many variables and prone to errors in practice [8]. One approach to improve scalability in high-dimensional settings is to group variables into clusters and infer relationships between these clusters. In the context of diagrams constructed from knowledge used for identification of causal effects, Cluster Directed Acyclic Graphs (C-DAGs) [1] are introduced as causal diagrams defined over clusters, allowing the visual representation of a high-dimensional system to be simplified and the requisite knowledge for graph specification lessened. In a C-DAG, nodes are clusters of variables, and an edge exists if a variable in one cluster causally influences a variable in another. C-DAGs are assumed to be constructed based on partial knowledge of causal and confounding relationships between variables across clusters, oblivious to variable-level relationships within clusters.

In this work, we address causal discovery over clusters of variables. We assume that the underlying causal model is a Markov DAG over individual variables $\mathbf{V} = \{V_1, ...V_n\}$ with no latent variables. Given a predefined partition of $\mathbf{V}$ into clusters $\mathbf{C} = \{\mathbf{C}_1, \ldots, \mathbf{C}_k\}$, we aim to learn causal relationships between these clusters based on observed conditional (in)dependencies between clusters encoded in the distribution $P(\mathbf{C}) = P(\mathbf{C}_1, \ldots, \mathbf{C}_k)$ without access to variable-level relationships.

One might attempt to simply treat each cluster as a multivariate random variable and apply existing causal discovery algorithms like PC [18]. However, consider the DAG $G_1$ and its corresponding C-DAG $G_{\mathbf{C}_1}$ in Figure 1(b) and 1(a), respectively. Assuming a probability distribution faithful to $G_1$, PC will correctly construct the skeleton $\mathbf{X} - \mathbf{Z} - \mathbf{Y}$, but observing the independence $\mathbf{X} \perp\!\!\!\perp \mathbf{Y}$ will lead to the collider structure $\mathcal{P}_{\mathbf{C}_1}$ in Figure 1(c), clearly misrepresenting the true causal directions. In fact, we have both $\mathbf{X} \perp\!\!\!\perp \mathbf{Y}$ and $\mathbf{X} \perp\!\!\!\perp \mathbf{Y}|\mathbf{Z}$ according to $G_1$. No DAG structures over clusters $\mathbf{X} - \mathbf{Z} - \mathbf{Y}$ can simultaneously capture both independencies. This implies the need for a new graphical object to represent (in)dependence information between clusters. Suppose we revise our collider test to only assign a collider to a triplet $\langle \mathbf{X}, \mathbf{Z}, \mathbf{Y} \rangle$ when $\mathbf{X} \perp\!\!\!\perp \mathbf{Y}$ and $\mathbf{X} \not\perp\!\!\!\perp \mathbf{Y}|\mathbf{Z}$. Consider $G_2$ and its C-DAG $G_{\mathbf{C}_2}$ in Figure 1(e), and 1(d), respectively. In this context, our modified collider test allows correct determination of the collider structure $\mathbf{X} \to \mathbf{Z} \leftarrow \mathbf{Y}$ (and no other colliders). Applying the standard orientation rule that for triplet $\mathbf{X} \to \mathbf{Z} - \mathbf{W}$, $\mathbf{Z} - \mathbf{W}$ should be oriented as $\mathbf{Z} \to \mathbf{W}$ to reflect that $\langle \mathbf{X}, \mathbf{Z}, \mathbf{W} \rangle$, not yet oriented, must be a non-collider again results in a misdirected edge.

These somewhat surprising results illustrate the complexities of representing causal and independence relationships over clusters and show that naively applying existing algorithms like PC over clusters can lead to incorrect orientations. PC over individual variables learns a Markov equivalence class of causal diagrams with the same conditional (in)dependencies [19, 20, 9, 22], represented as a completed partially directed acyclic graph (CPDAG) [6, 9, 2]. Analogously, for clusters, the goal is to recover a Markov equivalence class reflecting the same (in)dependencies between clusters.

**Summary of Contributions** Our contributions are as follows:

1. In section 2, we define a new graphical object, $\alpha$C-DAG (Definition 7), that, in addition to causal relations, explicitly represents all (in)dependence information over clusters. We define a new criterion for d-separation in $\alpha$C-DAGs (Definition 8) which we show is sound and complete for extracting conditional (in)dependencies over clusters (Theorem 1).

2. In section 3, we define *Cluster Completed Partially Directed Acyclic Graphs*, or $\alpha$C-CPDAGs, to represent a Markov equivalence class of $\alpha$C-DAGs (Definition 10). We

introduce a learning algorithm for sound and complete causal discovery over clusters to learn an $\alpha$C-CPDAG by testing conditional (in)dependencies over clusters (Algorithm 1).

## 1.1 Related work and Preliminaries

In the literature, clusters are mainly used as an intermediate step in learning a graphical equivalence class over variables. Typically, clusters of nodes sharing some properties are learned, then structures within or between these clusters are learned, and ultimately integrated into a graph over variables representing a class of DAGs [21, 12, 4, 5, 25]. Prior approaches that learn structures over clusters either group variables heuristically based on structural similarity [10], assume clusters with strict internal structural constraints [3, 16], including where structures such as those in Figure 1 are disallowed [11, 24], or consider only two clusters [23]. In contrast, we consider a user-defined partition of variables and learn a structure representing a cluster-level equivalence class.

**Notation.** A single variable is denoted by a (non-boldface) uppercase letter $X$ and its realized value by a small letter $x$. A boldfaced uppercase letter $\mathbf{X}$ denotes a set (or a cluster) of variables. We use kinship relations, defined via edges in the graph. We denote by $Pa(\mathbf{X})_G$, $Ch(\mathbf{X})_G$, $An(\mathbf{X})_G$, and $De(\mathbf{X})_G$, the sets of parents, children, ancestors, and descendants in graph $G$, respectively. A triplet $\langle V_i, V_k, V_j \rangle$ is *active* if 1) $V_k$ is a collider and $V_k$ or any of its descendants are in $\mathbf{Z}$ or 2) $V_k$ is a non-collider and is not in $\mathbf{Z}$. A path $p$ is said to be *active* given (or conditioned on) $\mathbf{Z}$ if every triplet on $p$ is active relative to $\mathbf{Z}$. Otherwise, $p$ is said to be *inactive*. Given a graph $G$, $\mathbf{X}$ and $\mathbf{Y}$ are d-separated by $\mathbf{Z}$ if every path between $\mathbf{X}$ and $\mathbf{Y}$ is inactive given $\mathbf{Z}$. We denote this d-separation by $(\mathbf{X} \perp\!\!\!\perp \mathbf{Y} \mid \mathbf{Z})_G$. **Learned Equivalence Classes.** A completed partially directed acyclic graph (CPDAG) $\mathcal{G}$ can have either directed ($\rightarrow$) or undirected ($-$) edges. Directed edges are common for all members of the Markov equivalence class represented by the CPDAG whereas undirected edges are variant. A triplet of vertices $\langle X, Y, Z \rangle$ is unshielded if $X$ and $Z$ are not adjacent to each other. If $X$ and $Z$ are adjacent to one another, the triplet is said to be shielded. In a consecutive triplet $\langle X, Z, Y \rangle$, $Z$ is a definite collider if edges from $X$ and $Y$ are into it ($X \rightarrow Z \leftarrow Y$). $Z$ is a definite non-collider if at least one edge is out of it ($X \leftarrow Z - Y$, $X - Z \rightarrow Y$) or both edges are undirected and the triplet is unshieleded ($X - Z - Y$). Otherwise, $Z$ has a non-definite status. **Cluster DAG or C-DAG (Markov)**[1] Given a DAG $G(\mathbf{V}, \mathbf{E})$ and a partition $\mathbf{C} = \{\mathbf{C}_1, \ldots, \mathbf{C}_k\}$ of $\mathbf{V}$, construct a graph $G_{\mathbf{C}}(\mathbf{C}, \mathbf{E}_{\mathbf{C}})$ over $\mathbf{C}$ with a set of edges $\mathbf{E}_{\mathbf{C}}$ defined as follows: An edge $\mathbf{C}_i \rightarrow \mathbf{C}_j$ is in $\mathbf{E}_{\mathbf{C}}$ if exists some $V_i \in \mathbf{C}_i$ and $V_j \in \mathbf{C}_j$ such that $V_i \in Pa(V_j)$ in $G$. If $G_{\mathbf{C}}(\mathbf{C}, \mathbf{E}_{\mathbf{C}})$ contains no cycles, then we say that $\mathbf{C}$ is an *admissible partition* of $\mathbf{V}$. We then call $G_{\mathbf{C}}$ a *cluster DAG*, or *C-DAG*, compatible with $G$. The definition of d-separation over C-DAGs extends from that over variables and is elaborated on in Appendix A and [1].

## 2 $\alpha$C-DAGs: a new graphical object for encoding causal relationships and (in)dependencies over clusters

### 2.1 Representing (in)dependence information over clusters

In DAGs, marginal and conditional (in)dependencies align consistently with structural edges and arrowhead orientations between variables. As d-separation rules familiarly show, for an unshielded triplet $X, Z, Y$, a collider structure exists if and only if $X \perp\!\!\!\perp Y$ and $X \not\perp\!\!\!\perp Y | Z$. A non-collider structure exists if and only if $X \not\perp\!\!\!\perp Y$ and $X \perp\!\!\!\perp Y | Z$. It is only possible for $X \not\perp\!\!\!\perp Y$ and $X \not\perp\!\!\!\perp Y | Z$ if the triplet is shielded. The last combination of independence information, $X \perp\!\!\!\perp Y$ and $X \perp\!\!\!\perp Y | Z$ such that $X$ and $Y$ are adjacent as well as $Z$ and $Y$, never occurs. With C-DAGs, ambiguity is introduced and the correspondence between graphical structure and independence information changes. Consider $G_1$ and $G_2$ in Figure 2(a), which are both colliders over the clusters $\langle \mathbf{X}, \mathbf{Z}, \mathbf{Y} \rangle$, but are each associated with distinct independence information. $G_3$ and $G_4$ illustrate analogous behavior for non-colliders, whether a chain or fork. Therefore, neither collider nor non-collider structures over clusters can be singularly associated with specific independencies or dependencies, unlike with variables. Fortunately, the converse is true: certain independence tests can singularly inform structure, and we can leverage this property for learning over clusters in some cases. However, a new representation is needed to ensure complete representation of independence information for structural inference.

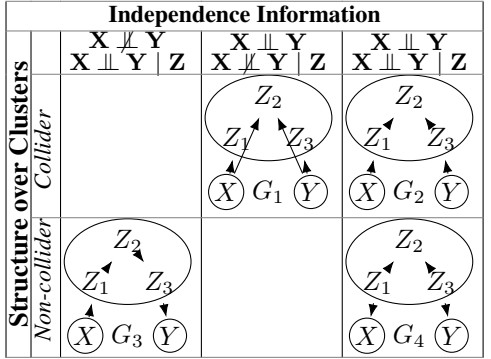

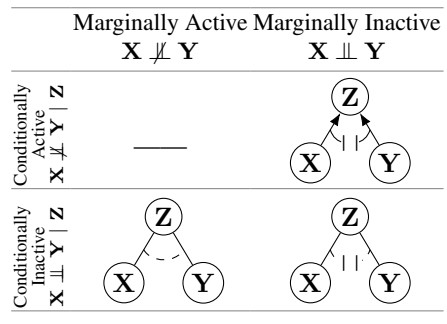

(a) Example DAGs representing non-colliders and colliders with possible independence information for clusters.

(b) Independence-arcs for marginal/ conditional independence/dependence combinations.

Figure 2: Graphical Structures and Representations of Independence Information.

## 2.2 A novel representation of independence information

We introduce a new semantic representation called "independence arcs" to graphically encode known independence information. These arcs explicitly convey independence information between variables, decoupled from ancestral relationships. We note that while the terms of "edges" and "arcs" are often used interchangeably to refer to the connections between nodes in a graph, we use the term "independence arc" to refer to a novel symbolic representation of an arc *drawn between two edges* of a cluster graph. The form and representation of the arc conveys information about the conditional and marginal (in)dependencies of the triplet of which these two edges are a part. This is in contrast to what we consistently refer to as *edges*, meaning the connections between nodes in a graph.

Figure 2(b) shows the three new independence arc markings and their meanings, defined formally in Definition 2. A break in the independence arc indicates a marginally inactive triplet, while an arc without any break represents a marginally active triplet. A dashed arc indicates a conditionally inactive triplet, while a solid line indicates a conditionally active triplet. Under this new representation, edges preserve their semantics with regards to conveying parent-child relationships between nodes, and independence information of a triplet is determined exclusively through the independence arc.

Independence arcs annotate both unshielded triplets, $\langle \mathbf{C}_i, \mathbf{C}_k, \mathbf{C}_j \rangle$, where $\mathbf{C}_k$ is adjacent to both $\mathbf{C}_i$ and $\mathbf{C}_j$, and $\mathbf{C}_i$ and $\mathbf{C}_j$ are not adjacent, and shielded triplets, $\langle \mathbf{C}'_i, \mathbf{C}'_k, \mathbf{C}'_j \rangle$, where $\mathbf{C}'_k$ is adjacent to both $\mathbf{C}'_i$ and $\mathbf{C}'_j$, and $\mathbf{C}'_i$ and $\mathbf{C}'_j$ are adjacent. To determine the arc for a shielded triplet, we introduce the concept of *manipulation of a shielded triplet* where one edge of the triplet is removed so that the triplet can become unshielded, and the arc describes the behavior of this induced unshielded triplet.

**Definition 1** (Manipulation of a shielded triplet). *Given a shielded triplet over clusters $\langle \mathbf{C}_i, \mathbf{C}_k, \mathbf{C}_j \rangle$, its manipulation involves removing the edge between $\mathbf{C}_i$ and $\mathbf{C}_j$, corresponding to removal of any edges between variables in these clusters. After manipulation, the shielded triplet becomes unshielded and this manipulated unshielded triplet is referenced as $\langle \mathbf{C}_i, \mathbf{C}_k, \mathbf{C}_j \rangle^{-\mathbf{C}_i \mathbf{C}_j}$.*

**Example 1:** Consider Figure 3. Triplet $\langle \mathbf{A}, \mathbf{B}, \mathbf{E} \rangle$ in $G_{\mathbf{C}_1}$ is shielded. To manipulate the triplet, the edge $\mathbf{A} \to \mathbf{E}$ is removed, corresponding to removing the edge $A_1 \to E_2$ in $G_1$. This manipulated unshielded triplet in $G_{\mathbf{C}_1}$ is referred to as $\langle \mathbf{A}, \mathbf{B}, \mathbf{E} \rangle^{-\mathbf{AE}}$. The complete process for adding independence arcs to a graph is described below in Definition 2.

**Definition 2** (**Independence Arcs**). *Consider a graph $\mathbf{G}_C$ over clusters $\mathbf{C} = \langle \mathbf{C}_0, ..., \mathbf{C}_n \rangle$. For any unshielded triplet $\langle \mathbf{C}_i, \mathbf{C}_k, \mathbf{C}_j \rangle$ (or manipulated unshielded triplet $\langle \mathbf{C}_i, \mathbf{C}_k, \mathbf{C}_j \rangle^{-\mathbf{C}_i \mathbf{C}_j}$), let $\mathbf{S}$ equal a (possibly empty) set of clusters $\mathbf{S} \subset (\mathbf{C} \setminus \{\mathbf{C}_i, \mathbf{C}_j\})$ such that $\mathbf{C}_i \perp\!\!\!\perp \mathbf{C}_j | \mathbf{S}$, if such a set exists. For a triplet $\langle \mathbf{C}_i, \mathbf{C}_k, \mathbf{C}_j \rangle$, an independence arc, $\mathcal{A}_{\mathbf{C}_i, \mathbf{C}_k, \mathbf{C}_j} \in \mathcal{A}$, can be drawn from some point on the edge between $\mathbf{C}_i$ and $\mathbf{C}_k$ to some point on the edge between $\mathbf{C}_j$ and $\mathbf{C}_k$ in the following way:*

*1. A marginally-connecting independence arc of - - - - is drawn if and only if $\mathbf{C}_k \in \mathbf{S}$. Consequently, $\mathbf{C}_i \not\perp\!\!\!\perp \mathbf{C}_j | \mathbf{S} \setminus \mathbf{C}_k$ and $\mathbf{C}_i \perp\!\!\!\perp \mathbf{C}_j | \mathbf{S}$.*

*2. A conditionally-connecting independence arc of —||— is drawn if and only if $\mathbf{C}_k \notin \mathbf{S}$ and $\mathbf{C}_i \not\perp\!\!\!\perp \mathbf{C}_j | \mathbf{S} \cup \mathbf{C}_k$.*

*3. A never-connecting independence arc of - -||- - is drawn if and only if $\mathbf{C}_k \notin \mathbf{S}$ and $\mathbf{C}_i \perp\!\!\!\perp \mathbf{C}_j | \mathbf{S} \cup \mathbf{C}_k$.*

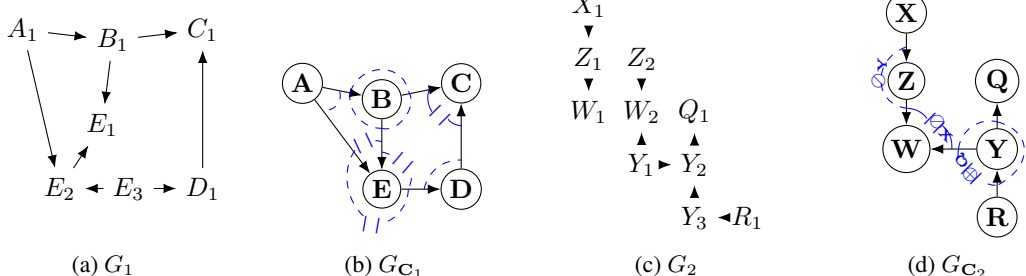

(a) $G_1$      (b) $G_{\mathbf{C}_1}$      (c) $G_2$      (d) $G_{\mathbf{C}_2}$

Figure 3: $G_1$ is a DAG in the class of $G_{\mathbf{C}_1}$, a C-DAG (with Independence Arcs). Independence arcs encode (in)dependencies between clusters, for example that $\mathbf{A} \perp\!\!\!\perp \mathbf{D}$ and $\mathbf{A} \not\perp\!\!\!\perp \mathbf{D}|\mathbf{C}$. $G_{\mathbf{C}_2}$ is a C-DAG (with Independence Arcs and Separation Marks, or $\alpha$C-DAG) and $G_2$ is a compatible DAG.

*Shielded triplets are annotated according to the behavior of their respective manipulated triplets.*

**Example 2:** Consider DAG $G_1$ in Figure 3. Unshielded triplets $\langle \mathbf{A}, \mathbf{B}, \mathbf{C} \rangle$, $\langle \mathbf{E}, \mathbf{B}, \mathbf{C} \rangle$, and $\langle \mathbf{C}, \mathbf{D}, \mathbf{E} \rangle$, are marked with a marginally-connecting arc, as are manipulated unshielded triplets $\langle \mathbf{E}, \mathbf{A}, \mathbf{B} \rangle^{-\mathbf{EB}}$ and $\langle \mathbf{A}, \mathbf{B}, \mathbf{E} \rangle^{-\mathbf{AE}}$. A conditionally-connecting arc is drawn for $\langle \mathbf{B}, \mathbf{C}, \mathbf{D} \rangle$. Never-connecting arcs are added to triplets $\langle \mathbf{A}, \mathbf{E}, \mathbf{D} \rangle$ and $\langle \mathbf{B}, \mathbf{E}, \mathbf{D} \rangle$, and manipulated unshielded triplet $\langle \mathbf{A}, \mathbf{E}, \mathbf{B} \rangle^{-\mathbf{AB}}$.

**Lemma 1.** *In a Markov C-DAG with independence arcs, a conditionally-connecting independence arc always implies a collider structure.*

While a collider structure $\mathbf{X} \to \mathbf{Z} \leftarrow \mathbf{Y}$ in a C-DAG does not necessarily imply that $\mathbf{X} \perp\!\!\!\perp \mathbf{Y}$ and $\mathbf{X} \not\perp\!\!\!\perp \mathbf{Y}|\mathbf{Z}$, lemma 1 notes that the converse is true. Independence arcs allow for d-separations to be read in a new way, unrelated to edge connections. For an isolated triplet with clusters $\langle \mathbf{C}_i, \mathbf{C}_k, \mathbf{C}_j \rangle$, the triplet is active (d-connecting) relative to the (possibly empty) set of cluster vertices $\mathbf{Z}$ if a) $\langle \mathbf{C}_i, \mathbf{C}_k, \mathbf{C}_j \rangle$ is marked with a marginally-connecting independence arc and $\mathbf{C}_k \notin \mathbf{Z}$ or b) $\langle \mathbf{C}_i, \mathbf{C}_k, \mathbf{C}_j \rangle$ is marked with a conditionally-connecting independence arc and $\mathbf{C}_k \in \mathbf{Z}$. Otherwise, $\langle \mathbf{C}_i, \mathbf{C}_k, \mathbf{C}_j \rangle$ is d-separated relative to $\mathbf{Z}$. In a larger graph, we introduce the notion of *arc trajectories*, or the sequence of independence arcs corresponding to a path between two variables. Arc trajectories can be analyzed to determine if two variables are connected or not.

**Definition 3** (**Arc Trajectory**). *Given a graph $\mathbf{G}_C$, for some path over clusters $\langle \mathbf{C}_1, \mathbf{C}_2, \mathbf{C}_3, ..., \mathbf{C}_n \rangle$, the* arc trajectory *refers to the sequence of independence arcs for each triplet along the path,* $\mathbf{a} = \langle \mathcal{A}_{\mathbf{C}_1, \mathbf{C}_2, \mathbf{C}_3}, ..., \mathcal{A}_{\mathbf{C}_{n-2}, \mathbf{C}_{n-1}, \mathbf{C}_n} \rangle$.

**Example 3:** Consider the example in Figure 3. To determine if $\mathbf{A}$ and $\mathbf{D}$ are d-separated ($\mathbf{A} \perp\!\!\!\perp \mathbf{D}$) in $G_{\mathbf{C}_1}$, we first identify all simple paths between $\mathbf{A}$ and $\mathbf{D}$, of which there are three: $\mathbf{A} \to \mathbf{B} \to \mathbf{C} \leftarrow \mathbf{D}$, $\mathbf{A} \to \mathbf{B} \to \mathbf{E} \to \mathbf{D}$, and $\mathbf{A} \to \mathbf{E} \to \mathbf{D}$. The arc trajectory corresponding to the first path is $\langle \mathcal{A}_{\mathbf{A}, \mathbf{B}, \mathbf{C}}, \mathcal{A}_{\mathbf{B}, \mathbf{C}, \mathbf{D}} \rangle$, consisting of a marginally-connecting arc and a conditionally-connecting arc. Because there is no conditioning set in the query, only $\mathcal{A}_{\mathbf{A}, \mathbf{B}, \mathbf{C}}$ indicates an active triplet but not $\mathcal{A}_{\mathbf{B}, \mathbf{C}, \mathbf{D}}$, and therefore $\mathbf{A}$ and $\mathbf{D}$ are not connected along this path. For the second path, the arc trajectory is $\langle \mathcal{A}_{\mathbf{A}, \mathbf{B}, \mathbf{E}}, \mathcal{A}_{\mathbf{B}, \mathbf{E}, \mathbf{D}} \rangle$. $\mathcal{A}_{\mathbf{A}, \mathbf{B}, \mathbf{E}}$ is a marginally-connecting arc, but $\mathcal{A}_{\mathbf{B}, \mathbf{E}, \mathbf{D}}$ is a never-connecting arc, so $\mathbf{A}$ and $\mathbf{D}$ are not connected by this path either. The last path has the arc trajectory $\langle \mathcal{A}_{\mathbf{A}, \mathbf{E}, \mathbf{D}} \rangle$, and its only independence arc is never-connecting. Therefore, we can conclude that $\mathbf{A} \perp\!\!\!\perp \mathbf{D}$. By a similar analysis, we can conclude that $\mathbf{A} \not\perp\!\!\!\perp \mathbf{D}|\mathbf{C}$.

With some simple examples, we illustrate that determining d-separations by independence arcs can sometimes be more complex. Consider Figure 3. From $G_2$, the following independence information is clear: $\mathbf{X} \not\perp\!\!\!\perp \mathbf{W}$ and $\mathcal{A}_{\mathbf{X}, \mathbf{Z}, \mathbf{W}}$ is a marginally-connecting arc, $\mathbf{Z} \not\perp\!\!\!\perp \mathbf{Y}|\mathbf{W}$, and $\mathcal{A}_{\mathbf{Z}, \mathbf{W}, \mathbf{Y}}$ is a conditionally-connecting arc. Then the arc trajectory in $G_{\mathbf{C}_2}$ from $\mathbf{X}$ to $\mathbf{Y}$ might lead us to believe that $\mathbf{X} \not\perp\!\!\!\perp \mathbf{Y}|\mathbf{W}$, but this is not true. Independence arcs indicate information with regards to a triplet of clusters, but alone, may misrepresent d-separation for paths over clusters. We enrich independence arcs with a new semantic representation to denote unexpected independencies. We introduce a new symbol, $\oslash_{\mathbf{C}}$, which we call a "separation mark." This mark annotates an independence arc of a triplet to indicate a cluster (specified by the subscript of the separation mark) further along on a path that, by independence arcs, would appear to have a d-connection to the variables in the triplet, but is actually separated. This notion is formalized in definition 5. First, we define a supporting concept below.

**Definition 4** (**Analogous Paths**). *Given a C-DAG $G_{\mathbf{C}}$ and a compatible DAG $G$, we define a simple path in $G$ over variables, $p = \langle V_1, V_2, V_3, ..., V_m \rangle$ to be considered analogous to a path in $G_{\mathbf{C}}$ over clusters $p_{\mathbf{C}} = \langle \mathbf{C}_1, \mathbf{C}_2, \mathbf{C}_3, ..., \mathbf{C}_n \rangle$ (and $p_{\mathbf{C}}$ analogous to $p$) if and only if the following hold: 1) for*

*every variable $V_i$ on $p$, $V_i$ is in some cluster $\mathbf{C}_i$ on $p_\mathbf{C}$, 2) for every cluster $C_j$ on $p_\mathbf{C}$, there exists some variable $V_j \in \mathbf{C}_j$ where $V_j$ is on $p$, and 3) for any variable $V_n \in \mathbf{C}_n$, there does not exist any variable that appears after $V_n$ on $p$ that is in a cluster before $\mathbf{C}_n$ on $p_\mathbf{C}$.*

In Fig. 3, the path over variables $p_v = \langle A_1, B_1, C_1, D_1 \rangle$ in $G_1$ is an analogous path for the path over clusters $p_c = \langle \mathbf{A}, \mathbf{B}, \mathbf{C}, \mathbf{D} \rangle$ in $G_{\mathbf{C}_1}$, but the path over variables $p'_v = \langle A_1, B_1, E_1, E_2, E_3, D_1 \rangle$ is not analogous to $p_c$, since $\mathbf{E}$ is not on $p_c$ but $\exists V_e \in \mathbf{E}$ on $p'_v$ and $\nexists V_c \in \mathbf{C}$ on $p'_v$, but $\mathbf{C}$ is on $p_c$.

**Definition 5 (Separation Marks).** *Let $G$ be a DAG, and let $G_\mathbf{C}$ denote a possible C-DAG for $G$. Consider a path $p_\mathbf{C}$ in $G_\mathbf{C}$ over clusters $\langle \mathbf{C}_1, \mathbf{C}_2, \mathbf{C}_3, ..., \mathbf{C}_n \rangle$ and its corresponding arc trajectory $\mathbf{a} = \langle \mathcal{A}_{\mathbf{C}_1, \mathbf{C}_2, \mathbf{C}_3}, ... \mathcal{A}_{\mathbf{C}_{n-2}, \mathbf{C}_{n-1}, \mathbf{C}_n} \rangle$ such that:*

1. *there is no arc $\mathcal{A}_{\mathbf{C}_i, \mathbf{C}_{i+1}, \mathbf{C}_{i+2}} \in \mathbf{a}$ that is a never-connecting arc,*
2. *there is no d-connecting path $p$ in $G$ over variables relative to some set of clusters $\mathbf{Z}$, analogous to $p_\mathbf{C}$,*
3. *there exists a d-connecting path $p'$ in $G$ over variables relative to some set of clusters $\mathbf{Z}'$ that is analogous to the path $p'_\mathbf{C} = \langle \mathbf{C}_1, ..., \mathbf{C}_{n-1} \rangle$ in $G_\mathbf{C}$, and*
4. *there exists a d-connecting path $p''$ in $G$ over variables relative to some set $\mathbf{Z}''$ of clusters that is analogous to the path $p''_\mathbf{C} = \langle \mathbf{C}_2, ..., \mathbf{C}_n \rangle$ in $G_\mathbf{C}$, .*

*Then, a separation mark, $\oslash_{\mathbf{C}_1}$ is placed on the arc $\mathcal{A}_{\mathbf{C}_{n-2}, \mathbf{C}_{n-1}, \mathbf{C}_n}$, and a separation mark, $\oslash_{\mathbf{C}_n}$ is placed on the arc $\mathcal{A}_{\mathbf{C}_1, \mathbf{C}_2, \mathbf{C}_3}$.*

**Example 4:** In Figure 3, we identify where a separation mark is needed by traversing paths of length greater than 3 in $G_{\mathbf{C}_2}$ and compare to the paths over variables in $G_2$. For example, traversing the path $\langle \mathbf{X}, \mathbf{Z}, \mathbf{W}, \mathbf{Y} \rangle$ in $G_{\mathbf{C}_2}$ and comparing to $G_2$, we see that there is no path between any variable in $\mathbf{X}$ and a variable in $\mathbf{Y}$. We place a separation mark with the subscript $\mathbf{Y}$, as in $\oslash_\mathbf{Y}$, on the independence arc of $\mathcal{A}_{\mathbf{X}, \mathbf{Z}, \mathbf{W}}$. This indicates that when traversing a path starting at $\mathbf{X}$ where $\mathcal{A}_{\mathbf{X}, \mathbf{Z}, \mathbf{W}}$ is in the arc trajectory associated with the path, $\mathbf{Y}$ is separated from $\mathbf{X}$ (in addition to any nodes past $\mathbf{Y}$ on the path). We place a mirroring separation mark, $\oslash_\mathbf{X}$, along arc trajectory $\mathcal{A}_{\mathbf{Z}, \mathbf{W}, \mathbf{Y}}$ to reflect the reverse. $G_{\mathbf{C}_2}$ in Figure 3 shows the C-DAG with independence arcs and separation marks. Further discussion on separation marks can be found in Appendix C.

Separation marks indicate separations on paths masked by the clusters and independence arcs. Connections may also be masked if conditioning on a descendant of a collider within a cluster, where the descendant is in a different cluster from the collider. We introduce a new *connection mark*, which, like separation marks, annotates independence arcs. Specifically, a connection mark, $\oplus_{\mathbf{C}_x}$ on an independence arc $\mathcal{A}_{\mathbf{C}_i, \mathbf{C}_k, \mathbf{C}_j}$ denotes that the triplet $\langle \mathbf{C}_i, \mathbf{C}_k, \mathbf{C}_j \rangle$ is activated by conditioning on $\mathbf{C}_x$ due to some variable $V_x \in \mathbf{C}_x$ being a descendant of some collider variable $V_k \in \mathbf{C}_k$.

**Definition 6 (Connection Marks).** *Let $G$ be a DAG and let $G_\mathbf{C}$ denote a possible C-DAG for $G$ with independence arcs. Consider a triplet over clusters in $G_\mathbf{C}$, $\langle \mathbf{C}_i, \mathbf{C}_k, \mathbf{C}_j \rangle$, and its corresponding independence arc, $\mathcal{A}_{\mathbf{C}_i, \mathbf{C}_k, \mathbf{C}_j}$. If $\mathcal{A}_{\mathbf{C}_i, \mathbf{C}_k, \mathbf{C}_j}$ is a never-connecting or conditionally-connecting independence arc, and there exists a path $p$ in $G$ over variables through the triplet $\langle V_i, ..., V_k, ... V_j \rangle$ such that $V_i \in \mathbf{C}_i$, $V_k \in \mathbf{C}_k$, and $V_j \in \mathbf{C}_j$ then $\forall V'_k \in \mathbf{C}_k$ and on $p$, where $V'_k$ is a collider, let $\mathbf{D}$ be the set of clusters that are children of $\mathbf{C}_k$ and which include descendants of all colliders along the path, $(\mathbf{D} = \bigcup \{ \mathbf{C}_d : V_d \in \mathbf{C}_d$ where $V_d \notin \{ \mathbf{C}_i, \mathbf{C}_k, \mathbf{C}_j \}$ and $V_d \in Ch(V_k))$. Then the connection mark $\oplus_\mathbf{D}$ is added to $\mathcal{A}_{\mathbf{C}_i, \mathbf{C}_k, \mathbf{C}_j}$.*

**Example 5:** Consider again Figure 3. Collider $Y_2$ in the triplet $\langle Y_1, Y_2, Y_3 \rangle$ in $G_2$ is not discernible in triplet $\langle \mathbf{W}, \mathbf{Y}, \mathbf{R} \rangle$ in $G_{\mathbf{C}_2}$, which is marked by a never-connecting independence arc. However, conditioning on $\mathbf{Q}$ renders $\mathbf{R}$ and $\mathbf{W}$ dependent. The connection mark $\oplus_\mathbf{Q}$ is placed along arc $\mathcal{A}_{\mathbf{W}, \mathbf{Y}, \mathbf{R}}$, as shown. Further discussion on connection marks can be found in Appendix C.

### 2.3 $\alpha$C-DAG Definition and Properties

With the introduction of the new symbolic representations of independence arcs, separation marks, and connection marks we can fully define a new graphical model for C-DAGs with independence arcs, which we call $\alpha$C-DAGs, for short. The "$\alpha$" prefix will be used to indicate graphical representations making use of the new semantics of independence arcs, separation marks and connection marks.

**Definition 7 ($\alpha$C-DAG (C-DAG with Independence Arcs)).** *Given a DAG $G(\mathbf{V}, \mathbf{E})$ and a partition $\mathbf{C} = \{ \mathbf{C}_1, ..., \mathbf{C}_n \}$ of $\mathbf{V}$, construct a graph $G_\mathbf{C}(\mathbf{C}, \mathbf{E}_\mathbf{C}, \mathcal{A})$ over $\mathbf{C}$.*

- *An edge $\mathbf{C}_i \to \mathbf{C}_j$ is in $\mathbf{E}_\mathbf{C}$ if exists some $V_i \in \mathbf{C}_i$ and $V_j \in \mathbf{C}_j$ such that $V_i \in Pa(V_j)$ in $G$;*

- *The set of independence arcs $\mathbf{A}$ is defined over all triplets $\langle \mathbf{C}_i, \mathbf{C}_k, \mathbf{C}_j \rangle$, by Definition 2.*
- *For each arc trajectory in $G_{\mathbf{C}}$, separation marks are added according to Definition 5.*
- *For each path in $G_{\mathbf{C}}$, connection marks are added according to Definition 6.*

*If for all pairs of clusters $\mathbf{C}_i, \mathbf{C}_j$ where there exists an edge $\mathbf{C}_i \rightarrow \mathbf{C}_j$, there is no directed path $\mathbf{C}_j \rightarrow ... \rightarrow \mathbf{C}_i$, then we say that $\mathbf{C}$ is an admissible partition of $\mathbf{V}$. We then call $G_{\mathbf{C}}$ a* cluster DAG with independence arcs, *or an $\alpha$C-DAG, compatible with $G$.*

As with the definition of C-DAGs, $\alpha$C-DAGs assume acyclicity over clusters. Specifically, we disallow what we define as **apparent directed cycles** (or just apparent cycles), where edges over clusters give the appearance of a cycle such that for some pair of clusters $\{\mathbf{C}_i, \mathbf{C}_j\}$ there exists an edge $\mathbf{C}_i \rightarrow \mathbf{C}_j$ and a directed path $\mathbf{C}_j \rightarrow ... \rightarrow \mathbf{C}_i$. While Definition 7 takes as input a DAG, we also note that construction of an $\alpha$C-DAG could alternatively take as input a C-DAG and a probability distribution $P(\mathbf{C})$ where $P(\mathbf{C})$ is faithful to the true data-generating process. Knowledge would inform edge directions and $P(\mathbf{C})$ would inform independence arcs, separation marks and connections marks; the $\alpha$C-DAG is still considered an object constructed from knowledge, rather than one that is learned.

D-separation over $\alpha$C-DAGs can be determined according to the criteria below. In the theorem that follows, we show these d-separation rules are sound and complete in $\alpha$C-DAGs.

**Definition 8** (**d-separation over $\alpha$C-DAGs.**). *A path $p_{\mathbf{C}}$ in an $\alpha$C-DAG, $G_{\mathbf{C}}$, is said to be d-separated (or blocked) by a set of clusters $\mathbf{Z} \subset \mathbf{C}$ if and only if its corresponding arc trajectory $\mathbf{a}$ contains an independence arc $\mathcal{A}_{\mathbf{C}_i, \mathbf{C}_k, \mathbf{C}_j}$ that is:*

1. *a marginally-connecting independence arc and (a) $\mathbf{C}_k$ is in $\mathbf{Z}$ or (b) there exists a separation mark $\oslash_{\mathbf{C}_x}$ on $\mathcal{A}_{\mathbf{C}_i, \mathbf{C}_k, \mathbf{C}_j}$ where $\mathbf{C}_x$ is on $p_{\mathbf{C}}$.*
2. *a conditionally-connecting independence arc and (a) $\mathbf{C}_k$ is not in $\mathbf{Z}$ nor is any true descendant $\mathbf{C}_d$ of $\mathbf{C}_k$ (with directed and connecting path $\mathbf{C}_k \rightarrow ... \rightarrow \mathbf{C}_d$) in $\mathbf{Z}$, and (b) for any connection mark $\oplus_{\mathbf{C}_x}$ on $\mathcal{A}_{\mathbf{C}_i, \mathbf{C}_k, \mathbf{C}_j}$, $\mathbf{C}_x$ is not in $\mathbf{Z}$ or (c) there exists a separation mark on $\mathcal{A}_{\mathbf{C}_i, \mathbf{C}_k, \mathbf{C}_j} \oslash_{\mathbf{C}_x}$ where $\mathbf{C}_x$ is on $p_{\mathbf{C}}$.*
3. *a never-connecting independence arc and for connection mark $\oplus_{\mathbf{C}_x}$ on $\mathcal{A}_{\mathbf{C}_i, \mathbf{C}_k, \mathbf{C}_j}$, $\mathbf{C}_x \notin \mathbf{Z}$.*

**Theorem 1.** *[Soundness and completeness of d-separation in $\alpha$C-DAGs.] In an $\alpha$C-DAG $G_{\mathbf{C}}$, let $\{\mathbf{X}, \mathbf{Z}, \mathbf{Y}\} \in \mathbf{C}$. $\mathbf{X}$ and $\mathbf{Y}$ are d-separated by $\mathbf{Z}$ in $G_{\mathbf{C}}$, if and only if for any DAG, $G$, compatible with $G_{\mathbf{C}}$, $\mathbf{X}$ and $\mathbf{Y}$ are d-separated by $\mathbf{Z}$ in $G$. $(\mathbf{X} \perp\!\!\!\perp \mathbf{Y} \mid \mathbf{Z})_{G_{\mathbf{C}}} \iff (\mathbf{X} \perp\!\!\!\perp \mathbf{Y} \mid \mathbf{Z})_G$.*

This d-separation definition informs how (in)dependencies can be read over clusters in an $\alpha$C-DAG. The novel graph of an $\alpha$C-DAG represents knowledge of both connections between and (in)dependence information over clusters. In the next section, we build on the $\alpha$-CDAG semantics introduced here to define a new graphical object foundational for learning over clusters.

## 3 $\alpha$C-CPDAGs and learning

### 3.1 Equivalence classes of $\alpha$C-DAGs

As with other causal discovery algorithms, our approach to learning over clusters will result in a graphical equivalence class of compatible models, specifically an equivalence class of $\alpha$C-DAGs. This equivalence class will represent the class of graphs, over clusters, that share the same independence structure induced, and the associated graphical object is analogous to a CPDAG, which uniquely represents a Markov equivalence class over variables. We define this novel graph as a cluster CPDAG, or $\alpha$C-CPDAG. In this section, we define the relationship of this object to $\alpha$C-DAGs and describe how $\alpha$C-CPDAGs can be learned from an observational distribution.

Two DAGs, $G_1$ and $G_2$ with the same vertices are Markov equivalent if for any three disjoint sets of vertices $\mathbf{X}, \mathbf{Z}, \mathbf{Y}$, $\mathbf{X}$ and $\mathbf{Y}$ are d-separated by $\mathbf{Z}$ in $G_1$ if and only if $\mathbf{X}$ and $\mathbf{Y}$ are d-separated by $\mathbf{Z}$ in $G_2$. We extend a similar notion for clusters and $\alpha$C-DAGs in Definition 9. From the definition of d-separation for $\alpha$C-DAGs, we know that such separations are discernible from independence arcs, separation marks and connection marks, which leads to the theorem following the definition. In $\alpha$C-CPDAGs, it may not always be possible to determine the independence arcs associated with each manipulated unshielded triplet within a shielded triplet. In this case, it is possible for any arc to exist, and all corresponding graphs are included in the equivalence class represented by the $\alpha$C-CPDAG.

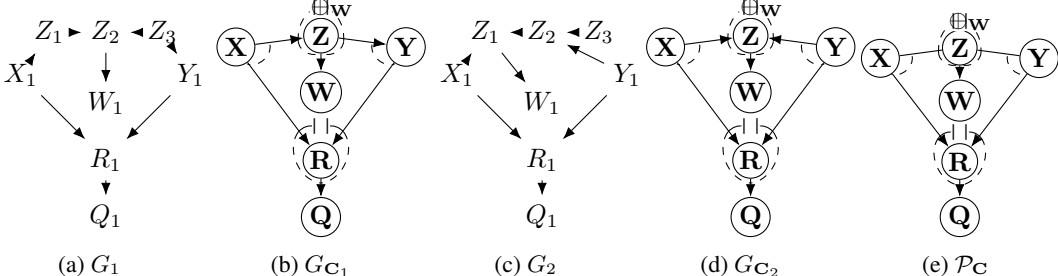

(a) $G_1$  (b) $G_{\mathbf{C}_1}$  (c) $G_2$  (d) $G_{\mathbf{C}_2}$  (e) $\mathcal{P}_{\mathbf{C}}$

Figure 4: $G_1$ and $G_2$ are DAGs in the classes of $\alpha$C-DAGs $G_{\mathbf{C}_1}$ and $G_{\mathbf{C}_2}$, respectively. $G_{\mathbf{C}_1}$ and $G_{\mathbf{C}_2}$ are in the Markov equivalence class of $\alpha$C-CPDAG, $\mathcal{P}_{\mathbf{C}}$. In $\mathcal{P}_{\mathbf{C}}$, $\mathscr{R}_0$ is applied to $\langle \mathbf{X}, \mathbf{R}, \mathbf{Y} \rangle$, and then $\mathscr{R}_1$ is applied to $\langle \mathbf{X}, \mathbf{R}, \mathbf{Q} \rangle$. Lastly, $\mathscr{R}_5$ is applied to $\langle \mathbf{X}, \mathbf{Z}, \mathbf{Y} \rangle$ with descendant $\mathbf{W}$.

**Definition 9** (**Cluster Markov Equivalence**). *Two $\alpha$C-DAGs, $G_{\mathbf{C}_1}$ and $G_{\mathbf{C}_2}$ (with the same partition $\mathbf{C}$ over the same variables $\mathbf{V}$) are cluster Markov equivalent if for any three disjoint sets of clusters $\mathbf{X}, \mathbf{Z}, \mathbf{Y}$, $\mathbf{X}$ and $\mathbf{Y}$ are d-separated by $\mathbf{Z}$ in $G_{\mathbf{C}_1}$ iff $\mathbf{X}$ and $\mathbf{Y}$ are d-separated by $\mathbf{Z}$ in $G_{\mathbf{C}_2}$.*

**Theorem 2.** *Two $\alpha$C-DAGs, $G_{\mathbf{C}_1}$ and $G_{\mathbf{C}_2}$ (with the same partition $\mathbf{C}$ over the same set of variables $\mathbf{V}$) are cluster Markov equivalent if and only if they share the same: 1) adjacencies, 2) independence arcs, 3) separation marks and 4) connection marks.*

Figure 4 illustrates example DAGs and $\alpha$C-DAGs in the same cluster Markov equivalence class. Markov equivalent $\alpha$C-DAGs share some unshielded colliders, namely those marked by a conditionally-connecting arc. This characterization of equivalent $\alpha$C-DAGs leads to the definition of the cluster CPDAGs ($\alpha$C-CPDAGs). As with a partially directed acyclic graph, an $\alpha$C-CPDAG may contain both directed and undirected edges and does not contain any directed cycles. As with $\alpha$C-DAGs, an $\alpha$C-CPDAG is defined over a user-defined partition of clusters $\mathbf{C}$ over the variables $\mathbf{V}$.

**Definition 10** ($\alpha$**Cluster CPDAG**). *Let $[G_{\mathbf{C}}]$ be the Markov equivalence class of an arbitrary $\alpha$C-DAG, $G_{\mathbf{C}}$. The cluster completed partially directed acyclic graph ($\alpha$C-CPDAG) for $[G_{\mathbf{C}}]$, denoted $\mathcal{P}$, is defined such that:*

1. *$\mathcal{P}$ has the same adjacencies as $G_{\mathbf{C}}$ (and therefore any member of $[G_{\mathbf{C}}]$).*
2. *A directed edge is in $\mathcal{P}$ iff shared by all $\alpha$C-DAGs in $[G_{\mathbf{C}}]$; otherwise the edge is undirected.*
3. *An independence arc is in $\mathcal{P}$ iff shared by all $\alpha$C-DAGs in $[G_{\mathbf{C}}]$; otherwise there is no arc.*
4. *$\mathcal{P}$ has the same separation and connection marks as $G_{\mathbf{C}}$ (and any member of $[G_{\mathbf{C}}]$).*

### 3.2 A Constraint-Based Learning Algorithm for $\alpha$C-CPDAGs

Given definitions of the relationships between an $\alpha$C-DAG and a DAG, and an $\alpha$C-CPDAG and an $\alpha$C-DAG, we can develop an approach for the reverse process of constructing an $\alpha$C-CPDAG from (in)dependencies in an observational dataset. Algorithm 1, Causal Learning Over Clusters (CLOC), defines this procedure. CLOC assumes that an available distribution $P(\mathbf{C})$ (or data representing it) is *faithful* to the true $\alpha$C-DAG (see definition 11) and that partition $\mathbf{C}$ is admissible. Figure 4 illustrates an $\alpha$C-CPDAG learned by the algorithm.

**Definition 11** (**Faithfulness for $\alpha$C-DAGs**). *Given an $\alpha$C-DAG, $G_{\mathbf{C}}$, and probability distribution over the clusters, $P(\mathbf{C})$, that is generated by an SCM consistent with any causal diagram compatible with $G_{\mathbf{C}}$, we say that $P(\mathbf{C})$ is* faithful *to $G_{\mathbf{C}}$ if $(\mathbf{X} \perp\!\!\!\perp \mathbf{Y}|\mathbf{Z})_{P(\mathbf{c})} \Rightarrow (\mathbf{X} \perp\!\!\!\perp \mathbf{Y}|\mathbf{Z})_{P(G_{\mathbf{C}})}$.*

CLOC has three phases. In the first, edges between pairs of nodes, $\mathbf{X}$ and $\mathbf{Y}$, are removed from a complete graph with undirected edges if there exists some separating set of clusters $\mathbf{S}$ such that $(\mathbf{X} \perp\!\!\!\perp \mathbf{Y}|\mathbf{S})$. Independence arcs are added and colliders are determined from conditionally-connecting arcs (Lemma 1). In the second phase, separation marks, and connection marks are added. In the final phase, five orientation rules are evaluated until none apply. Rules 1, 3 and 4 extend from PC, leveraging independence arcs to determine where the logic is sound. Rule 2 extends precisely, and Rule 5 is our contribution. This algorithm gives us an $\alpha$C-CPDAG, which represents the cluster Markov equivalence class of $\alpha$C-DAGs compatible with the distribution $P(\mathbf{C})$. We review the rules below with proofs in Appendix B. After, we demonstrate that the orientation rules and the learning algorithm are sound and complete for learning causal relations between clusters. Note that in the orientation rules, asterisks indicate either an arrowhead or tail is possible.

$\mathscr{R}_0$: If $\mathbf{X} - \mathbf{Z} - \mathbf{Y}$ and $\mathcal{A}_{\mathbf{X},\mathbf{Z},\mathbf{Y}}$ is conditionally-connecting, then orient the triplet as $\mathbf{X} \to \mathbf{Z} \leftarrow \mathbf{Y}$

$\mathscr{R}_1$: If $\mathbf{X} \to \mathbf{Z} - \mathbf{Y}$, $\mathbf{X}$ and $\mathbf{Y}$ are not adjacent, and $\mathcal{A}_{\mathbf{X},\mathbf{Z},\mathbf{Y}}$ is marginally-connecting, then orient the triplet as $\mathbf{X} \to \mathbf{Z} \to \mathbf{Y}$.

$\mathscr{R}_2$: If $\mathbf{X} \to \mathbf{Z} \to \mathbf{Y}$ and $\mathbf{X} - \mathbf{Y}$, then orient $\mathbf{X} - \mathbf{Y}$ as $\mathbf{X} \to \mathbf{Y}$.

---

**Algorithm 1:** CLOC: Algorithm for Learning an $\alpha$C-CPDAG

**Input:** Admissible partition $\mathbf{C} = \{\mathbf{C}_1, ..., \mathbf{C}_n\}$, $P(\mathbf{C})$
**Output:** $\alpha$C-CPDAG, $\mathcal{P}$
    1. Determine skeleton, separation sets, independence arcs, and identifiable colliders by Algorithm 2.
    2. Add the separation and connection marks by Algorithm 3.
    3. Apply the five orientation rules until none apply.

---

**Algorithm 2:** CLOC: Adjacencies and Independence Arcs

**Input:** Partition $\mathbf{C} = \{\mathbf{C}_1, ..., \mathbf{C}_n\}$
**Output:** $\alpha$C-CPDAG, $\mathcal{P}$

1 Form complete graph $\mathcal{P}$ over $\mathbf{C}$ with undirected edges.
2 **for** $\mathbf{X}, \mathbf{Y} \in \mathbf{C}$ **do**
3     **for** $\mathbf{S} \subseteq \mathbf{C} \setminus \{\mathbf{X}, \mathbf{Y}\}$ **do**
4         **if** $P(\mathbf{y} \mid \mathbf{s}, \mathbf{x}) = P(\mathbf{y} \mid \mathbf{s})$ **then**
5             SepSet $\leftarrow \mathbf{S}$, SepFlag $\leftarrow True$, **break**
6     **if** $SepFlag = True$ **then**
7         Remove the edge between $\mathbf{X}, \mathbf{Y}$ in $\mathcal{P}$
8 **for** *every unshielded triplet* $\langle \mathbf{X}, \mathbf{Z}, \mathbf{Y} \rangle$ *in* $\mathcal{P}$ **do**
9     **if** $\mathbf{Z} \notin SepSet(\mathbf{X}, \mathbf{Y})$ *and* $\mathbf{X} \not\perp\!\!\!\perp \mathbf{Y} \mid \mathbf{Z} \cup SepSet(\mathbf{X}, \mathbf{Y})$ **then**
10         Mark $\langle \mathbf{X}, \mathbf{Z}, \mathbf{Y} \rangle$ in $\mathcal{P}$ with a conditionally-connecting arc, and orient as $\mathbf{X} \to \mathbf{Z} \leftarrow \mathbf{Y}$
11     **else if** $\mathbf{Z} \in SepSet(\mathbf{X}, \mathbf{Y})$ **then**
12         Mark $\langle \mathbf{X}, \mathbf{Z}, \mathbf{Y} \rangle$ in $\mathcal{P}$ with a marginally-connecting arc
13     **else**
14         Mark $\langle \mathbf{X}, \mathbf{Z}, \mathbf{Y} \rangle$ in $\mathcal{P}$ with a never-connecting arc

15 **for** *every shielded triplet* $\langle \mathbf{X}, \mathbf{Z}, \mathbf{Y} \rangle$ *in* $\mathcal{P}$ *with* $\mathbf{Y}$ *adjacent to some* $\mathbf{W}$ *and* $\mathbf{X}$ *not adjacent to* $\mathbf{W}$ **do**
16     **if** $\mathcal{A}_{\mathbf{Z},\mathbf{Y},\mathbf{W}}$ *is conditionally-connecting and* $\mathcal{A}_{\mathbf{X},\mathbf{Y},\mathbf{W}}$ *is not* **then**
17         **if** $\mathbf{Z} \in SepSet(\mathbf{X}, \mathbf{W})$ **then**
18             Mark $\langle \mathbf{X}, \mathbf{Z}, \mathbf{Y} \rangle^{-\mathbf{XY}}$ in $\mathcal{P}$ with a marginally-connecting arc
19         **else**
20             Mark $\langle \mathbf{X}, \mathbf{Z}, \mathbf{Y} \rangle^{-\mathbf{XY}}$ in $\mathcal{P}$ with a never-connecting arc

21     **else if** $\mathcal{A}_{\mathbf{Z},\mathbf{Y},\mathbf{W}}$ *is marginally-connecting and* $\mathcal{A}_{\mathbf{X},\mathbf{Y},\mathbf{W}}$ *is not* **then**
22         **if** $\mathbf{Z} \notin SepSet(\mathbf{X}, \mathbf{W})$ *and* $\mathbf{X} \not\perp\!\!\!\perp \mathbf{W} \mid \mathbf{Z} \cup SepSet(\mathbf{X}, \mathbf{Y})$ **then**
23             Mark $\langle \mathbf{X}, \mathbf{Z}, \mathbf{Y} \rangle^{-\mathbf{XY}}$ in $\mathcal{P}$ with a conditionally-connecting arc, and orient $\mathbf{X} \to \mathbf{Z} \leftarrow \mathbf{Y}$
24         **else if** $\mathbf{Z} \in SepSet(\mathbf{X}, \mathbf{W})$ **then**
25             Mark $\langle \mathbf{X}, \mathbf{Z}, \mathbf{Y} \rangle^{-\mathbf{XY}}$ in $\mathcal{P}$ with a marginally-connecting arc
26         **else**
27             Mark $\langle \mathbf{X}, \mathbf{Z}, \mathbf{Y} \rangle^{-\mathbf{XY}}$ in $\mathcal{P}$ with a never-connecting arc

---

**Algorithm 3:** CLOC: Separation and Connection Marks

**Input:** Admissible partition $\mathbf{C} = \{\mathbf{C}_1, ..., \mathbf{C}_n\}$, $P(\mathbf{C})$
**Output:** $\alpha$C-CPDAG, $\mathcal{P}$

1 **for** *each path* $p = \langle \mathbf{C}_0, ..., \mathbf{C}_n \rangle \in \mathcal{P}$ **do**
2     **if** *length*$(p) \geq 4$ *and arc trajectory* $\mathbf{a}$ *for* $p$ *is only marginal/conditionally-connecting arcs (with no marks* $\oslash_{\mathbf{C}_a}$ *where* $\mathbf{C}_a \in p$*)* **then**
3         Let $\mathbf{K} \leftarrow \bigcup \{C_z \mid \mathcal{A}_{C_x, C_z, C_y} \in \mathbf{a}$ is conditionally-connecting$\}$
4         **if** $\mathbf{C}_0 \perp\!\!\!\perp \mathbf{C}_n \mid \mathbf{K} \cup (SepSet(\mathbf{C}_0, \mathbf{C}_n) \setminus p)$ **then**
5             For shortest subpath $p' = \langle \mathbf{C}_i, ..., \mathbf{C}_j \rangle \subseteq p$ s.t. length$(p') \geq 4$ and $\mathbf{C}_i \perp\!\!\!\perp \mathbf{C}_j \mid \mathbf{K} \cup (SepSet(\mathbf{C}_0, \mathbf{C}_n) \setminus p)$
6             Add $\oslash_{\mathbf{C}_j}$ to $\mathcal{A}_{\mathbf{C}_i, \mathbf{C}_{i+1}, \mathbf{C}_{i+2}}$
7             Add $\oslash_{\mathbf{C}_i}$ to $\mathcal{A}_{\mathbf{C}_{j-2}, \mathbf{C}_{j-1}, \mathbf{C}_j}$

8 **for** *each triplet* $\langle \mathbf{X}, \mathbf{Z}, \mathbf{Y} \rangle$ *in* $\mathcal{P}$ **do**
9     **if** $\mathcal{A}_{\mathbf{X},\mathbf{Z},\mathbf{Y}}$ *is a conditionally or never connecting arc and* $\exists$ *some* $\mathbf{W}$ *such that* $\mathbf{Z} - \mathbf{W}$, $\mathbf{X}$ *and* $\mathbf{W}$ *are not adjacent,* $\mathbf{Y}$ *and* $\mathbf{W}$ *are not adjacent* **then**
10         Let $\mathcal{W}_{\mathbf{Z}} \leftarrow \{\mathbf{W} \mid \mathbf{Z} - \mathbf{W}$ exists in $\mathcal{P}\}$
11         Let $\mathcal{S}$ be the power set of $\mathcal{W}_{\mathbf{Z}} \setminus \emptyset$
12         **for** *each subset* $\mathcal{D} \in \mathcal{S}$ **do**
13             **if** $\mathbf{X} \not\perp\!\!\!\perp \mathbf{Y} \mid \mathcal{D} \cup SepSet(\mathbf{X}, \mathbf{Y})$ **then**
14                 Add $\oplus_{\mathcal{D}}$ to $\mathcal{A}_{\mathbf{X},\mathbf{Z},\mathbf{Y}}$

---

$\mathscr{R}_3$: If $\mathbf{X} \to \mathbf{Z} \leftarrow \mathbf{Y}$, $\mathbf{X} - \mathbf{W} - \mathbf{Y}$, $\mathbf{X}$ and $\mathbf{Y}$ are not adjacent, $\mathbf{W} - \mathbf{Z}$, and $\mathcal{A}_{\mathbf{X},\mathbf{W},\mathbf{Y}}$ is marginally-connecting, then orient $\mathbf{W} - \mathbf{Z}$ as $\mathbf{W} \to \mathbf{Z}$.

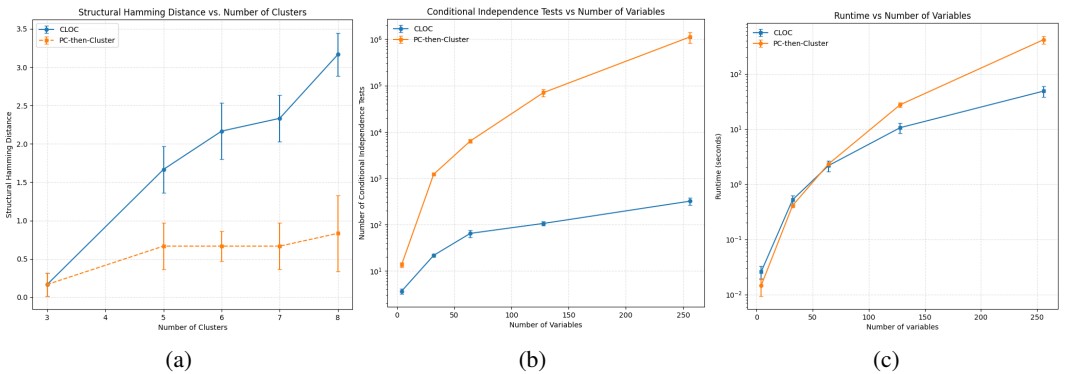

Figure 5: Plots comparing the (a) Structural Hamming Distance, (b) runtime, and (c) number of conditional-independence tests for CLOC (blue) compared to the PC-then-cluster approach (orange).

$\mathscr{R}_4$: If $\mathbf{X} \to \mathbf{Z} \to \mathbf{Y}$, $\mathbf{X} - \mathbf{W} - \mathbf{Y}$, $\mathbf{X}$ and $\mathbf{Y}$ are not adjacent, $\mathbf{W} \ast\!\!-\!\!\ast \mathbf{Z}$, and $\mathcal{A}_{\mathbf{X},\mathbf{W},\mathbf{Y}}$ is marginally-connecting, then orient $\mathbf{W} - \mathbf{Y}$ as $\mathbf{W} \to \mathbf{Y}$.

$\mathscr{R}_5$: If $\mathbf{X} \ast\!\!-\!\!\ast \mathbf{Z} \ast\!\!-\!\!\ast \mathbf{Y}$, $\mathbf{Z} - \mathbf{W}$, $\mathbf{X}$ and $\mathbf{W}$ are not adjacent, $\mathbf{Y}$ and $\mathbf{W}$ are not adjacent, and $\mathcal{A}_{\mathbf{X},\mathbf{Z},\mathbf{Y}}$ is never-connecting or conditionally-connecting with connection mark $\oplus_{\mathcal{D}}$ such that $\mathbf{W} \in \mathcal{D}$, then orient $\mathbf{Z} - \mathbf{W}$ as $\mathbf{Z} \to \mathbf{W}$.

**Theorem 3.** *[**Soundness and Completeness of Orientation Rules and CLOC**] The five orientation rules and the procedure of CLOC are sound and complete.*

## 4 Experiments

We show performance of CLOC in comparison to a 'PC-then-Cluster' approach, where PC over the entire set of variables is run with variables then grouped by the partition, yielding a graph over clusters. We generate random C-DAGs (3, 5, 6, 7, 8 clusters), and random DAGs (4, 8, 32, 64, 128, 256 variables) compatible with the C-DAGs. A Gaussian distribution (1000, 3000, 10000, 30000 samples) faithful to the DAG is drawn, over which PC-then-Cluster and CLOC are run. Runtime, conditional independence test counts, and structural hamming distances between the resulting graphs of each method and the true C-DAG are shown in Figure 5. Design details, implementation code, and additional results are included in Appendix D. PC requires exponentially more independence tests relative to CLOC. Runtime is also improved for CLOC. More efficient multivariate independence tests can lead to greater runtime improvements for CLOC. The structural hamming distances of the graphs generated by PC and CLOC differ in expected ways (see discussion in Appendix D).

## 5 Conclusions

In this work, we address the need for causal discovery over clusters in Markov causal systems. We introduce $\alpha$C-DAGs, which capture both causal directions and (in)dependence information over clusters, setting the stage for introduction of $\alpha$C-CPDAGs, an equivalence class of $\alpha$C-DAGs and the graphical object representing the class of cluster graphs with shared (in)dependencies and orientations. We then propose a sound and complete algorithm, CLOC, to learn this new graphical equivalence class from data, capturing much of the information that could be learned from variables, with fewer independence tests and faster runtime. Limitations of the approach include assumptions of causal sufficiency and faithfulness which may not apply for all applications. Users are required to have knowledge of a partition of variables into clusters that does not induce a cycle, which is non-negligible, while feasible for many applications including in clinical and biological domains, where partitions often arise naturally and may correspond to laboratory panels, gene sets, microbiome groups, neuroanatomical regions, or demographic blocks. While tests over clusters may have lower statistical power or be slower than those over individual variables, this can be effectively mitigated by advances in multivariate testing (e.g., Mantel Test), and modern machine learning-based methods that reliably assess independence between multivariate distributions. The foundational work introduced here sets the stage for improved scalability and makes possible causal discovery for sets of variables.

## Acknowledgements

This research is supported in part by the NSF, ONR, AFOSR, DoE, Amazon, JP Morgan, The Alfred P. Sloan Foundation, the United States National Library of Medicine grants (T15LM007079, R01LM006910).

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

# List of Appendices

## A  Additional Background

### A.1  d-Separation in C-DAGs

The definition of d-Separation in C-DAGs, introduced in [1], is provided below.

**Definition 12** (**D-Separation in C-DAGs**). *A path $p$ in a C-DAG $G_{\mathbf{C}}$ is said to be d-separated (or blocked) by a set of clusters $\mathbf{Z} \subset \mathbf{C}$ if and only if $p$ contains a triplet*

    *1. $\mathbf{C}_i \ast\!\!-\!\!\ast \mathbf{C}_m \to \mathbf{C}_j$ such that the non-collider cluster $\mathbf{C}_m$ is in $\mathbf{Z}$, or*
    *2. $\mathbf{C}_i \ast\!\!\to \mathbf{C}_m \leftarrow\!\ast \mathbf{C}_j$ such that the collider cluster $\mathbf{C}_m$ and its descendants are not in $\mathbf{Z}$.*

*A set of clusters $\mathbf{Z}$ is said to d-separate two sets of clusters $\mathbf{X}, \mathbf{Y} \subset \mathbf{C}$, denoted by $(\mathbf{X} \perp\!\!\!\perp \mathbf{Y} \mid \mathbf{Z})_{G_{\mathbf{C}}}$, if and only if $\mathbf{Z}$ blocks every path from a cluster in $\mathbf{X}$ to a cluster in $\mathbf{Y}$.*

This definition, in the context of a triplet $\langle \mathbf{C}_i, \mathbf{C}_m, \mathbf{C}_j \rangle$ reflects that $\mathbf{C}_i$ and $\mathbf{C}_j$ are d-separated by $\mathbf{C}_m$ if and only if all paths over variable through the clusters of the triplet are d-separated by the set of variables in cluster $\mathbf{C}_m$.

## B  Proofs

**Lemma 1.** *In a Markov C-DAG with independence arcs, a conditionally-connecting independence arc always implies a collider structure.*

*Proof.* Consider an unshielded triplet $\langle \mathbf{C}_i, \mathbf{C}_k, \mathbf{C}_j \rangle$ such that $\mathcal{A}_{\mathbf{C}_i, \mathbf{C}_k, \mathbf{C}_j}$ is a conditionally-connecting independence arc. This implies that $\mathbf{C}_i \perp\!\!\!\perp \mathbf{C}_j | \mathbf{S} \setminus \mathbf{C}_k; \mathbf{C}_i \not\perp\!\!\!\perp \mathbf{C}_j | \mathbf{C}_k \cup \mathbf{S}$ where $\mathbf{S}$ is a separating set for $\mathbf{C}_i$ and $\mathbf{C}_j$. Then there must exist some path, $p = V_i, ..., V_k, ...V_j$ where $V_i \in \mathbf{C}_i$, $V_k \in \mathbf{C}_k$, and $V_j \in \mathbf{C}_j$, such that every non-endpoint node is a collider. In Markovian cases, this can only occur if there is only one non-endpoint. Therefore, $V_k$ must be the only non-endpoint node on $p$ such that $V_k$ is a collider. Moreover, due to the admissibility of the partition, it follows that no additional variable in $\mathbf{C}_k$ can act as a cause of any variable in $\mathbf{C}_i$ or $\mathbf{C}_j$. Therefore, $langle\mathbf{C}_i, \mathbf{C}_k, \mathbf{C}_j \rangle$ must follow a collider structure. $\square$

**Theorem 1.** *[Soundness and completeness of d-separation in $\alpha$C-DAGs.] In an $\alpha$C-DAG $G_{\mathbf{C}}$, let $\{\mathbf{X}, \mathbf{Z}, \mathbf{Y}\} \in \mathbf{C}$. $\mathbf{X}$ and $\mathbf{Y}$ are d-separated by $\mathbf{Z}$ in $G_{\mathbf{C}}$, if and only if for any DAG, $G$, compatible with $G_{\mathbf{C}}$, $\mathbf{X}$ and $\mathbf{Y}$ are d-separated by $\mathbf{Z}$ in $G$. $(\mathbf{X} \perp\!\!\!\perp \mathbf{Y} \mid \mathbf{Z})_{G_{\mathbf{C}}} \iff (\mathbf{X} \perp\!\!\!\perp \mathbf{Y} \mid \mathbf{Z})_G$.*

*Proof.* First we prove the soundness of d-separation by showing that if $\mathbf{X}$ and $\mathbf{Y}$ are d-separated by $\mathbf{Z}$ in $G_{\mathbf{C}}$, then, in any DAG, $G$, compatible with $G_{\mathbf{C}}$, $\mathbf{X}$ and $\mathbf{Y}$ are d-separated by $\mathbf{Z}$ in $G$. We show by contradiction. Assume $\mathbf{X}$ and $\mathbf{Y}$ are d-separated by $\mathbf{Z}$ in $G_{\mathbf{C}}$ but in some compatible DAG, $G$, there exists a path $p$ between a variable $X \in \mathbf{X}$ and $Y \in \mathbf{Y}$ that is active when the set of variables contained in cluster $\mathbf{Z}$ are conditioned on. By the preservation of paths and adjacencies, no connection is destroyed through clustering, so $p$ in $G$ is contained in a path $p_{\mathbf{C}}$ of $G_{\mathbf{C}}$ between clusters $\mathbf{X}$ and $\mathbf{Y}$. Since $\mathbf{X}$ and $\mathbf{Y}$ are d-separated by $\mathbf{Z}$ in $G_{\mathbf{C}}$, $p_{\mathbf{C}}$ is blocked, and $\mathbf{X}$ and $\mathbf{Y}$ are not adjacent. Therefore, by definition 8, there is at least one triplet of clusters in $p_{\mathbf{C}}$ that indicates a block on the path. Let this triplet be $\langle \mathbf{C}_i, \mathbf{C}_m, \mathbf{C}_j \rangle$, and let its associated independence arc be $\mathcal{A}_{\mathbf{C}_i, \mathbf{C}_m, \mathbf{C}_j}$ where $\mathbf{C}_m$ is distinct from $\mathbf{X}$ and $\mathbf{Y}$. Consider the subpath $p_{ij}$ of $p$ contained in the triplet $\langle \mathbf{C}_i, \mathbf{C}_m, \mathbf{C}_j \rangle$ in $p_{\mathbf{C}}$. Since $p$ is active by assumption, every subpath of $p$ is active, including $p_{ij}$. The triplet $\langle \mathbf{C}_i, \mathbf{C}_m, \mathbf{C}_j \rangle$ indicates a block on the path either if 1) $\mathcal{A}_{\mathbf{C}_i, \mathbf{C}_m, \mathbf{C}_j}$ is a never connecting arc with no connection marks $\oplus_{\mathbf{C}_d}$ such that $\mathbf{C}_d \in \mathbf{Z}$, 2) if $\mathcal{A}_{\mathbf{C}_i, \mathbf{C}_m, \mathbf{C}_j}$ is a marginally-connecting arc where $\mathbf{C}_m \in \mathbf{Z}$, 3) if $\mathcal{A}_{\mathbf{C}_i, \mathbf{C}_m, \mathbf{C}_j}$ is a conditionally-connecting arc such that $\mathbf{C}_m \notin \mathbf{Z}$ and with no connection mark $\oplus_{\mathbf{C}_d}$ such that $\mathbf{C}_d \notin \mathbf{Z}$ or 4) if there is a separation mark $\oslash_{\mathbf{C}_x}$ on $\mathcal{A}_{\mathbf{C}_i, \mathbf{C}_m, \mathbf{C}_j}$ such that $\mathbf{C}_x$ is on $p_{\mathbf{C}}$. In case 1, $p_{ij}$ cannot be a connecting path or a collider path so $p_{ij}$ would be inactive. In case 2, $p_{ij}$ cannot be a collider path, and since $\mathbf{C}_m \in \mathbf{Z}$, $p_{ij}$ cannot be active. In case 3, $p_{ij}$ cannot be a connecting path and since $\mathbf{C}_m \notin \mathbf{Z}$ and for any connection mark $\oplus_{\mathbf{C}_d}$, $\mathbf{C}_d \notin \mathbf{Z}$, $p_{ij}$ cannot be active. In case 4, definition 5 states that if $\mathcal{A}_{\mathbf{C}_i, \mathbf{C}_m, \mathbf{C}_j}$ is a marginally-connecting arc such that $\mathbf{C}_m \notin \mathbf{Z}$, or if $\mathcal{A}_{\mathbf{C}_i, \mathbf{C}_m, \mathbf{C}_j}$ is a conditionally-connecting arc such that $\mathbf{C}_m \in \mathbf{Z}$, then $p_{ij}$ may be active, but since $\mathcal{A}_{\mathbf{C}_i, \mathbf{C}_m, \mathbf{C}_j}$ is marked with a separation mark $\oslash_{\mathbf{C}_x}$, there must exist some sub-path $p_{ix}$ of $p$ from some $V_i \in \mathbf{C}_i$ to some $V_x \in \mathbf{C}_x$ such that $\mathbf{C}_x$ is on $p_{\mathbf{C}}$ that is inactive. Therefore, $p$ must be inactive, there is a contradiction, and we conclude that if $\mathbf{X}$ and $\mathbf{Y}$ are d-separated by $\mathbf{Z}$ in $G_{\mathbf{C}}$, then, in any DAG, $G$, compatible with $G_{\mathbf{C}}$, $\mathbf{X}$ and $\mathbf{Y}$ are d-separated by $\mathbf{Z}$ in $G$.

Then, we prove the completeness of d-separation by showing that if $\mathbf{X}$ and $\mathbf{Y}$ are d-separated by $\mathbf{Z}$ in a DAG $G$, then $\mathbf{X}$ and $\mathbf{Y}$ are d-separated by $\mathbf{Z}$ in a compatible $\alpha$C-DAG $G_{\mathbf{C}}$. We prove by contradiction. Assume all paths from some $X \in \mathbf{X}$ to some $Y \in \mathbf{Y}$ are blocked by $\mathbf{Z}$ in some DAG $G$, but $\mathbf{X}$ and $\mathbf{Y}$ are not d-separated by $\mathbf{Z}$ in $G_{\mathbf{C}}$, i.e. $(\mathbf{X} \not\perp \mathbf{Y} | \mathbf{Z})_{G_{\mathbf{C}}}$. If all paths from any $X \in \mathbf{X}$ to any $Y \in \mathbf{Y}$ are inactive by $\mathbf{Z}$, then by preservation of paths and adjacencies, $\mathbf{X}$ and $\mathbf{Y}$ are not adjacent in $G_{\mathbf{C}}$. No connections are destroyed through clustering so any $p$ in $G$ is contained in a path $p_{\mathbf{C}}$ of $G_{\mathbf{C}}$ between clusters $\mathbf{X}$ and $\mathbf{Y}$. Because $\mathbf{X} \not\perp \mathbf{Y} | \mathbf{Z}$ in $G_{\mathbf{C}}$, by Definition 8, there must exist some path $p_{\mathbf{C}}$ such that 1) for any triplet $\langle \mathbf{C}_i, \mathbf{C}_m, \mathbf{C}_j \rangle$ on $p_{\mathbf{C}}$, the independence arc $\mathcal{A}_{\mathbf{C}_i, \mathbf{C}_m, \mathbf{C}_j}$ marking it must not be marked by a separation mark $\oslash_{\mathbf{C}_k}$ where $\mathbf{C}_k$ is on $p_{\mathbf{C}}$, 2) for all marginally-connecting arcs $\mathbf{C}_m \notin \mathbf{Z}$, 3) for all conditionally connecting arcs $\mathbf{C}_m \in \mathbf{Z}$, or $\mathcal{A}_{\mathbf{C}_i, \mathbf{C}_m, \mathbf{C}_j}$ is marked with a connection mark $\oplus_{\mathbf{C}_d}$ and $\mathbf{C}_d$ or a true descendant is in $\mathbf{Z}$, 4) for all never-connecting arcs, $\mathcal{A}_{\mathbf{C}_i, \mathbf{C}_m, \mathbf{C}_j}$ is marked by a connection mark $\oplus_{\mathbf{C}_d}$ and $\mathbf{C}_d$ or a true descendant is in $\mathbf{Z}$.

For all paths $p$ from some $X \in \mathbf{X}$ to some $Y \in \mathbf{Y}$ in $G$ to be blocked, there must exist at least one triplet, $\langle V_i, V_m, V_j \rangle$, contained either within 1 cluster (i.e. $\langle V_i, V_m, V_j \rangle \in \mathbf{C}_m$) or between 2 (i.e. $\langle V_i, V_m \rangle \in \mathbf{C}_m, V_j \in \mathbf{C}_j$ or $V_i \in \mathbf{C}_i, \langle V_m, V_j \rangle \in \mathbf{C}_m$) or 3 clusters (i.e. $V_i \in \mathbf{C}_i, V_m \in \mathbf{C}_m, V_j \in \mathbf{C}_j$) on $p_{\mathbf{C}}$, that is blocked.

1. If the blocked triplet is a non-collider, $V_i \leftarrow V_m \rightarrow V_j$ or $V_i \rightarrow V_m \rightarrow V_j$, then $V_m$ must be in $\mathbf{Z}$, which implies that $\mathbf{C}_m \in \mathbf{Z}$. As there could be multiple paths through a cluster, the triplet over clusters, $\langle \mathbf{C}_i, \mathbf{C}_m, \mathbf{C}_j \rangle$ could still be marked by any independence arc.
   (a) If $\mathcal{A}_{\mathbf{C}_i, \mathbf{C}_m, \mathbf{C}_j}$ is a marginally-connecting arc or never-connecting arc, since $\mathbf{C}_m \in \mathbf{Z}$, there is a contradiction with the implications of $(\mathbf{X} \not\perp \mathbf{Y} | \mathbf{Z})_{G_{\mathbf{C}}}$.
   (b) If $\mathcal{A}_{\mathbf{C}_i, \mathbf{C}_m, \mathbf{C}_j}$ is a conditionally-connecting arc, then then there must exist a different path, $p'$, over variables through the triplet from some some $V_i' \in \mathbf{C}_i$ to $V_j' \in \mathbf{C}_j$ through $\mathbf{C}_m$ that is a collider path. Because $\mathbf{C}_m \in \mathbf{Z}$, either there is no $X \in \mathbf{X}$ or $Y \in \mathbf{Y}$ on $p'$ or there must be another triplet, $V_q, V_r, V_w$, on $p'$ that is blocked.
2. If the triplet is a collider, $V_i \rightarrow V_m \leftarrow V_j$, then $V_m$ nor any of its descendants, $V_d$ can be in $\mathbf{Z}$, implying that $C_m \notin \mathbf{Z}$ and $\mathbf{C}_d \notin \mathbf{Z}$ where $V_d \in \mathbf{C}_d$ and $\mathcal{A}_{\mathbf{C}_i, \mathbf{C}_m, \mathbf{C}_j}$ is marked with the connection mark $\oplus_{\mathbf{C}_d}$.
   (a) If $\mathcal{A}_{\mathbf{C}_i, \mathbf{C}_m, \mathbf{C}_j}$ is a marginally-connecting arc, then there must exist a different path, $p'$, over variables through the triplet from some some $V_i' \in \mathbf{C}_i$ to $V_j' \in \mathbf{C}_j$ through $\mathbf{C}_m$

that is a connecting path. Because $\mathbf{C}_m \notin \mathbf{Z}$, either there is no $X \in \mathbf{X}$ or $Y \in \mathbf{Y}$ on $p'$ or there must be another triplet, $V_q, V_r, V_w$, on $p'$ that is blocked.

(b) If $\mathcal{A}_{\mathbf{C}_i, \mathbf{C}_m, \mathbf{C}_j}$ is a conditionally-connecting arc or a never-connecting arc, because $\mathbf{C}_m \notin \mathbf{Z}$, and there is a connection mark $\oplus_{\mathbf{C}_d}$, $\mathbf{C}_d \notin \mathbf{Z}$, there is a contradiction with the implications of $(\mathbf{X} \not\perp \mathbf{Y} | \mathbf{Z})_{G_\mathbf{C}}$.

For any path $p'$ with a blocked triplet $\langle V_q, V_r, V_w \rangle$, either one of the conditions above leading to a contradiction (case 1a or 2b) applies, or there is a contradiction because a separation mark must exist along the path $p_\mathbf{C}$. By definition 5, the separation mark would be required because by assumption, all paths between any $X \in \mathbf{X}$ and $Y \in \mathbf{Y}$ are blocked by $\mathbf{Z}$ in $G$, so it is not possible for there to be a d-connecting path relative to $\mathbf{Z}$ in $G$ analogous to $p_\mathbf{C}$ in $\mathbf{G}_C$. However, $p$ is a d-connecting path relative to $\mathbf{Z}$ analogous to $p'_\mathbf{C} = \langle \mathbf{C}_i, ..., \mathbf{C}_r \rangle$ and $p'$ is a d-connecting path relative to $\mathbf{Z}$ analogous to $p''_\mathbf{C} = \langle \mathbf{C}_m, ..., \mathbf{C}_w \rangle$, so by definition 5, the criteria is met and a separation must be placed.

If $\mathbf{X}$ and $\mathbf{Y}$ are d-separated by $\mathbf{Z}$ in $G$, it is also possible that there is no path from any $X \in \mathbf{X}$ to any $Y \in \mathbf{Y}$, and $\mathbf{Z}$ would equal the empty set. In this case, by preservation of adjacencies, for any triplet $\langle \mathbf{C}_i, \mathbf{C}_m, \mathbf{C}_j \rangle$ along $p_\mathbf{C}$, there must be some $V_i \in \mathbf{C}_i$ adjacent to some $V_m \in \mathbf{C}_m$, and some $V'_m \in \mathbf{C}_m$ adjacent to some $V_j \in \mathbf{C}_j$. Then, there must exist some such triplet where $V_m$ is not adjacent to $V'_m$. If for all $V_m$ and $V'_m$ in $C_m$, $V_m$ and $V'_m$ are not adjacent, then $\mathcal{A}_{\mathbf{C}_i, \mathbf{C}_m, \mathbf{C}_j}$ must be marked with a never-connecting arc in $G_\mathbf{C}$ with no connection mark, and there would be a contradiction with the implications of $(\mathbf{X} \not\perp \mathbf{Y} | \mathbf{Z})_{G_\mathbf{C}}$. Otherwise, because $\mathbf{X}$ and $\mathbf{Y}$ are d-separated by $\mathbf{Z}$ in $G$, there must exist some connecting subpaths of $p_\mathbf{C}$, $\mathbf{C}_i, ..., \mathbf{C}_n$ and $\mathbf{C}_{i+1}, ..., \mathbf{C}_n + 1$ such that $\mathbf{C}_i \perp\!\!\!\perp \mathbf{C}_{n+1}$, which, by definition 5, necessitates a separation mark and then there would be a contradiction with the implications of $(\mathbf{X} \not\perp \mathbf{Y} | \mathbf{Z})_{G_\mathbf{C}}$. $\qquad\square$

**Theorem 2.** *Two $\alpha$C-DAGs, $G_{\mathbf{C}_1}$ and $G_{\mathbf{C}_2}$ (with the same partition $\mathbf{C}$ over the same set of variables $\mathbf{V}$) are cluster Markov equivalent if and only if they share the same: 1) adjacencies, 2) independence arcs, 3) separation marks and 4) connection marks.*

*Proof.* The proof follows directly from the definitions of cluster Markov equivalence, and d-separation for $\alpha$C-DAGs. Because d-separation is determined solely by the independence arcs, separation marks, and connection marks in a graph for a series of adjacent clusters, two $\alpha$C-DAGs with the same adjacencies, independence arcs, separation marks, and connection marks will necessarily lead to the same d-separations between clusters and will therefore be cluster Markov equivalent. $\qquad\square$

**Theorem 3.** *[**Soundness and Completeness of Orientation Rules and CLOC**] The five orientation rules and the procedure of CLOC are sound and complete.*

First we prove the completeness of the arc assignment procedure, the soundness of the collider search and each of the five orientation rules. We then establish orientation completeness by showing that, whenever no more rules can be applied, there exist two Markov-equivalent $\alpha$C-DAGs that differ in orientation of any undirected edge. The proof for the soundness and completeness of CLOC follows. First, we introduce two remarks complementing lemma 1, and an additional associated lemma.

**Remark 1.** *In a Markov C-DAG with independence arcs, a marginally-connecting independence arc always implies a non-collider structure.*

*Proof.* We prove by contradiction. Consider an unshielded triplet $\langle \mathbf{C}_i, \mathbf{C}_k, \mathbf{C}_j \rangle$ such that $\mathcal{A}_{\mathbf{C}_i, \mathbf{C}_k, \mathbf{C}_j}$ is a marginally-connecting independence arc. We show that orienting the triple as $\mathbf{C}_i \to \mathbf{C}_k \leftarrow \mathbf{C}_j$ necessarily leads to a contradiction. By definition of a marginally-connecting independence arc, we have $\mathbf{C}_i \not\perp \mathbf{C}_j | \mathbf{S} \setminus \mathbf{C}_k; \mathbf{C}_i \perp\!\!\!\perp \mathbf{C}_j | \mathbf{S} \cup \mathbf{C}_k$, where $\mathbf{S}$ is a separating set for $\mathbf{C}_i$ and $\mathbf{C}_j$. Assume that the structure over clusters forms a collider, $\mathbf{C}_i \to \mathbf{C}_k \leftarrow \mathbf{C}_j$. There are two possible cases: either there is no path at all between $\mathbf{C}_i$ and $\mathbf{C}_j$ through $\mathbf{C}_k$, or such a path exists. If no such path exists, then the dependence implied by the marginally-connecting independence arc $\mathcal{A}_{\mathbf{C}_i, \mathbf{C}_k, \mathbf{C}_j}$ cannot hold, leading to a contradiction. If there exists a path $p$ between $\mathbf{C}_i$ and $\mathbf{C}_j$ through $\mathbf{C}_k$, then, since $\mathbf{C}_i$ is assumed to point to $\mathbf{C}_k$, there must be a pair of nodes $V_i \in \mathbf{C}_i$ and $V_k \in \mathbf{C}_k$ on $p$ such that $V_i \to V_k$. By the admissibility of the partition, an edge of the form $V_i \leftarrow V_k$ is not allowed. To preserve the marginal dependence implied by the marginally-connecting independence arc $\mathcal{A}_{\mathbf{C}_i, \mathbf{C}_k, \mathbf{C}_j}$, every subsequent edge between $V_k, V_{k+1} \in \mathbf{C}_k$ along the path $p$ must be of the form $V_k \to V_{k+1}$. Otherwise, a collider would be introduced, rendering the path inactive and violating the assumed marginal dependence, leading to a contradiction. Now, because $\mathbf{C}_k \leftarrow \mathbf{C}_j$, there must also exist some $V_j \in \mathbf{C}_j$ and some

$V'_k \in \mathbf{C}_k$ such that $V'_k \leftarrow V_j$ where $V'_k$ is on $p$. Because of the assumption of the admissibility of the partition, there can be no edge $V'k \rightarrow V_j$. Then there must exist a collider and there is a contradiction. Therefore, the triplet $\langle \mathbf{C}_i, \mathbf{C}_k, \mathbf{C}_j \rangle$ must be a non-collider. $\square$

**Remark 2.** *In a Markov C-DAG with independence arcs, a never-connecting independence arc could imply either a collider or a non-collider structure.*

*Proof.* Consider a triplet $\langle \mathbf{C}_i, \mathbf{C}_k, \mathbf{C}_j \rangle$ such that $\mathcal{A}_{\mathbf{C}_i, \mathbf{C}_k, \mathbf{C}_j}$ is a never-connecting independence arc. This implies that $\mathbf{C}_i \perp\!\!\!\perp \mathbf{C}_j | \mathbf{S} \setminus \mathbf{C}_k; \mathbf{C}_i \perp\!\!\!\perp \mathbf{C}_j | \mathbf{S} \cup \mathbf{C}_k$, where $\mathbf{S}$ is a separating set for $\mathbf{C}_i$ and $\mathbf{C}_j$. Then either there is no path from any $V_i \in \mathbf{C}_i$ to some $V_j \in \mathbf{C}_j$ through $\mathbf{C}_k$, or every such path $p$ must include at least 4 nodes, $p = V_i, ..., V_{k_1}, V_{k_2}, ..., V_j$ where $V_i \in \mathbf{C}_i$, $V_{k_1}, V_{k_2}, \in \mathbf{C}_k$, and $V_j \in \mathbf{C}_j$, such that there is at least one collider triplet and at least one non-collider triplet on $p$. Consider the latter case. Let $p$ be a path of exactly 4 nodes $\langle V_i, V_{k_1}, V_{k_2}, V_j \rangle$ such that $V_i \in \mathbf{C}_i$, $V_{k_1}, V_{k_2}, \in \mathbf{C}_k$ and $V_j \in \mathbf{C}_j$. Either $V_{k_1}$ is a collider node and $V_{k_2}$ is a non-collider node or $V_{k_1}$ is a non-collider node and $V_{k_2}$ is a collider node. In the first case, $V_i \rightarrow V_{k_1} \leftarrow V_{k_2} \rightarrow V_j$ or $V_i \rightarrow V_{k_1} \leftarrow V_{k_2} \leftarrow V_j$. In the second case, $V_i \rightarrow V_{k_1} \rightarrow V_{k_2} \leftarrow V_j$ or $V_i \leftarrow V_{k_1} \rightarrow V_{k_2} \leftarrow V_j$. Then $\langle \mathbf{C}_i, \mathbf{C}_k, \mathbf{C}_j \rangle$ may be either a collider or a non-collider. Adding any additional node, $V_{k_i+1}$, to $p$ either creates an additional collider or an additional non-collider, but still allows for collider and non-collider structures over clusters. Now consider where there is no path from any $V_i \in \mathbf{C}_i$ to some $V_j \in \mathbf{C}_j$ through $\mathbf{C}_k$. Then the direction of any edge $V_i - V_k$ or $V'_k - V_j$ can be variant such that $\langle \mathbf{C}_i, \mathbf{C}_k, \mathbf{C}_j \rangle$ may be either a collider or a non-collider. $\square$

**Lemma 2.** *For a distribution $P(\mathbf{C})$ over clusters $\mathbf{C} = \langle \mathbf{C}_1, ..., \mathbf{C}_n \rangle$ such that for every triplet $\langle \mathbf{C}_i, \mathbf{C}_k, \mathbf{C}_j \rangle$, $\mathcal{A}_{\mathbf{C}_i, \mathbf{C}_k, \mathbf{C}_j}$ is not a never-connecting independence arc, the orientation rules reduces to Meek's rules [9] and the PC algorithm [19].*

*Proof.* The proof follows from noting that modifications to Rules 1 and 3 require independence arcs aligning with the independence information typically associated with colliders and non-colliders over variables, and from lemma 1, and remarks 1, and 2. The absence of never-connecting arcs ensure triplets exhibit expected behavior with regards to structure and observed independencies and dependencies. When there are no never-connecting arcs, Rule 5 reduces to Rule 1, as all triplets marked with conditionally-connecting arcs must be a collider, and any descendant of that collider is part of a non-collider triplet, so will be oriented by Rule 1. When there are no never-connecting arcs and there is no background knowledge, Rule 4 never applies, following from Meek, 1995 [9]. $\square$

The procedure in Algorithm 2 for assigning independence arcs is sound and complete. The procedure for assigning independence arcs to unshielded triplets follows directly from the definitions of the arcs. We show that the procedure for identifying arcs for manipulated unshielded triplets is sound and complete, below.

*Proof.* Consider a shielded triplet $\langle \mathbf{X}, \mathbf{Z}, \mathbf{Y} \rangle$ and manipulated unshielded triplet $\langle \mathbf{X}, \mathbf{Z}, \mathbf{Y} \rangle^{-\mathbf{XY}}$. In isolation, no independence arc can be assigned to $\langle \mathbf{X}, \mathbf{Z}, \mathbf{Y} \rangle^{-\mathbf{XY}}$ as the information flow through the manipulated unshielded triplet cannot be isolated from edge $\mathbf{X} - \mathbf{Y}$. Therefore, to determine an arc for a manipulated unshielded triplet, at least one more node must be connected to the corresponding shielded triplet. Call this node $\mathbf{W}$ such that $\mathbf{W}$ is adjacent to $\mathbf{Y}$, $\mathbf{Y} - \mathbf{W}$ and $\mathbf{W}$ is not adjacent to $\mathbf{X}$. With this structure, there are two paths between $\mathbf{X}$ and $\mathbf{W}$. Let $p_1 = \langle \mathbf{X}, \mathbf{Y}, \mathbf{W} \rangle$ and let $p_2 = \langle \mathbf{X}, \mathbf{Z}, \mathbf{Y}, \mathbf{W} \rangle$. If independence arcs $\mathcal{A}_{\mathbf{X}, \mathbf{Y}, \mathbf{W}}$ and $\mathcal{A}_{\mathbf{Z}, \mathbf{Y}, \mathbf{W}}$ exist, we show that the independence arc for $\langle \mathbf{X}, \mathbf{Z}, \mathbf{Y} \rangle^{-\mathbf{XY}}$ can be determined if and only if $\mathcal{A}_{\mathbf{Z}, \mathbf{Y}, \mathbf{W}}$ is conditionally-connecting and $\mathcal{A}_{\mathbf{X}, \mathbf{Y}, \mathbf{W}}$ is not, or if $\mathcal{A}_{\mathbf{Z}, \mathbf{Y}, \mathbf{W}}$ is marginally-connecting and $\mathcal{A}_{\mathbf{X}, \mathbf{Y}, \mathbf{W}}$ is not.

If $\mathcal{A}_{\mathbf{Z}, \mathbf{Y}, \mathbf{W}}$ is conditionally-connecting and $\mathcal{A}_{\mathbf{X}, \mathbf{Y}, \mathbf{W}}$ is not, then $\mathbf{Z} \rightarrow \mathbf{Y}$ and $\mathbf{W} \rightarrow \mathbf{Y}$ by lemma 1, and $\mathcal{A}_{\mathbf{X}, \mathbf{Y}, \mathbf{W}}$ is either marginally-connecting or never-connecting. If $\mathcal{A}_{\mathbf{X}, \mathbf{Y}, \mathbf{W}}$ is marginally-connecting or never-connecting, then $p_1$ is blocked conditional on $\mathbf{Y}$. On $p_2$, triplet $\langle \mathbf{Z}, \mathbf{Y}, \mathbf{W} \rangle$ is active when conditioning on $\mathbf{Y}$. Then, if $\mathbf{X} \not\perp\!\!\!\perp \mathbf{W} | \mathbf{Y}$, $\mathcal{A}_{\mathbf{X}, \mathbf{Z}, \mathbf{Y}}$ must be marginally-connecting. If $\mathbf{X} \perp\!\!\!\perp \mathbf{W} | \mathbf{Y}$, $\mathcal{A}_{\mathbf{X}, \mathbf{Z}, \mathbf{Y}}$ must be never-connecting, because if $\mathcal{A}_{\mathbf{X}, \mathbf{Z}, \mathbf{Y}}$ were conditionally-connecting, then $\mathbf{Y} \rightarrow \mathbf{Z}$ and there would be a contradiction.

If $\mathcal{A}_{\mathbf{Z}, \mathbf{Y}, \mathbf{W}}$ is marginally-connecting and $\mathcal{A}_{\mathbf{X}, \mathbf{Y}, \mathbf{W}}$ is not, then $\mathcal{A}_{\mathbf{X}, \mathbf{Y}, \mathbf{W}}$ is either conditionally-connecting or never-connecting. If $\mathcal{A}_{\mathbf{X}, \mathbf{Y}, \mathbf{W}}$ is conditionally-connecting or never-connecting, then $p_1$ is blocked with no conditioning set. On $p_2$, triplet $\langle \mathbf{Z}, \mathbf{Y}, \mathbf{W}$ is active with no conditioning

set. Then, if $\mathbf{X} \not\perp\!\!\!\perp \mathbf{W}$, $\mathcal{A}_{\mathbf{X},\mathbf{Z},\mathbf{Y}}$ must be marginally-connecting. If $\mathbf{X} \not\perp\!\!\!\perp \mathbf{W}|\mathbf{Z}$, $\mathcal{A}_{\mathbf{X},\mathbf{Z},\mathbf{Y}}$ must be conditionally-connecting, and if $\mathbf{X} \perp\!\!\!\perp \mathbf{W}|\mathbf{Z}$, $\mathcal{A}_{\mathbf{X},\mathbf{Z},\mathbf{Y}}$ must be never-connecting,

Therefore the independence arc for $\langle \mathbf{X}, \mathbf{Z}, \mathbf{Y} \rangle^{-\mathbf{XY}}$ can be determined if $\mathcal{A}_{\mathbf{Z},\mathbf{Y},\mathbf{W}}$ is conditionally-connecting and $\mathcal{A}_{\mathbf{X},\mathbf{Y},\mathbf{W}}$ is not, or if $\mathcal{A}_{\mathbf{Z},\mathbf{Y},\mathbf{W}}$ is marginally-connecting and $\mathcal{A}_{\mathbf{X},\mathbf{Y},\mathbf{W}}$ is not. We now show that when these criteria do not hold, it impossible to determine the independence arc for $\langle \mathbf{X}, \mathbf{Z}, \mathbf{Y} \rangle^{-\mathbf{XY}}$. If $\mathcal{A}_{\mathbf{Z},\mathbf{Y},\mathbf{W}}$ is never-connecting, then, whether $\mathcal{A}_{\mathbf{X},\mathbf{Z},\mathbf{Y}}$ is marginally-connecting, conditionally-connecting, or never-connecting, $p_2$ will always be inactive. Therefore $\mathcal{A}_{\mathbf{Z},\mathbf{Y},\mathbf{W}}$ cannot be determined. If $\mathcal{A}_{\mathbf{Z},\mathbf{Y},\mathbf{W}}$ and $\mathcal{A}_{\mathbf{X},\mathbf{Y},\mathbf{W}}$ are conditionally-connecting, then to isolate $p_2$, $\mathbf{Y}$ should not be conditioned on so that $p_1$ is inactive, but then $p_2$ will also be blocked. If $\mathcal{A}_{\mathbf{Z},\mathbf{Y},\mathbf{W}}$ and $\mathcal{A}_{\mathbf{X},\mathbf{Y},\mathbf{W}}$ are marginally-connecting, then to isolate $p_2$, $\mathbf{Y}$ needs to be conditioned on to block $p_1$, but then $p_2$ will also be blocked. whether $\mathcal{A}_{\mathbf{X},\mathbf{Z},\mathbf{Y}}$ is marginally-connecting, conditionally-connecting, or never-connecting, $p_2$ will always be inactive. Therefore $\mathcal{A}_{\mathbf{Z},\mathbf{Y},\mathbf{W}}$ cannot be determined.

$\square$

$\mathscr{R}_0$: If $\mathbf{X} - \mathbf{Z} - \mathbf{Y}$ and $\mathcal{A}_{\mathbf{X},\mathbf{Z},\mathbf{Y}}$ is conditionally-connecting, then orient the triplet as $\mathbf{X} \to \mathbf{Z} \leftarrow \mathbf{Y}$

*Proof.* The proof of soundness follows directly from lemma 1.

$\square$

$\mathscr{R}_1$: If $\mathbf{X} \to \mathbf{Z} - \mathbf{Y}$, $\mathbf{X}$ and $\mathbf{Y}$ are not adjacent, and $\mathcal{A}_{\mathbf{X},\mathbf{Z},\mathbf{Y}}$ is marginally-connecting, then orient the triplet as $\mathbf{X} \to \mathbf{Z} \to \mathbf{Y}$.

*Proof.* The proof for soundness follows directly from Remark 1.

$\square$

$\mathscr{R}_2$: If $\mathbf{X} \to \mathbf{Z} \to \mathbf{Y}$ and $\mathbf{X} - \mathbf{Y}$, then orient $\mathbf{X} - \mathbf{Y}$ as $\mathbf{X} \to \mathbf{Y}$.

*Proof.* The soundness of the rule comes from observing that if $\mathbf{X} \leftarrow \mathbf{Y}$, a cycle would be induced, violating the admissible partition criteria of $\alpha$C-DAGs. $\square$

$\mathscr{R}_3$: If $\mathbf{X} \to \mathbf{Z} \leftarrow \mathbf{Y}$, $\mathbf{X} - \mathbf{W} - \mathbf{Y}$, $\mathbf{X}$ and $\mathbf{Y}$ are not adjacent, $\mathbf{W} - \mathbf{Z}$, and $\mathcal{A}_{\mathbf{X},\mathbf{W},\mathbf{Y}}$ is marginally-connecting, then orient $\mathbf{W} - \mathbf{Z}$ as $\mathbf{W} \to \mathbf{Z}$.

*Proof.* The soundness of the rule comes from observing that if $\mathbf{W} \leftarrow \mathbf{Z}$, then by two applications of rule 2, $\mathbf{Y} \to \mathbf{W}$, $\mathbf{X} \to \mathbf{W}$, and then there would be a collider at $\mathbf{W}$. Since $\mathcal{A}_{\mathbf{X},\mathbf{W},\mathbf{Y}}$ is marginally connecting, there is a contradiction by remark 1.

$\square$

$\mathscr{R}_4$: If $\mathbf{X} \to \mathbf{Z} \to \mathbf{Y}$, $\mathbf{X} - \mathbf{W} - \mathbf{Y}$, $\mathbf{X}$ and $\mathbf{Y}$ are not adjacent, $\mathbf{W} *\!\!-\!\!* \mathbf{Z}$, and $\mathcal{A}_{\mathbf{X},\mathbf{W},\mathbf{Y}}$ is marginally-connecting, then orient $\mathbf{W} - \mathbf{Y}$ as $\mathbf{W} \to \mathbf{Y}$.

*Proof.* The soundness of the rule comes from observing that if $\mathbf{W} \leftarrow \mathbf{Y}$, then to avoid a cycle, it must be that $\mathbf{X} \to \mathbf{W}$. Then, however, there would be a collider at $\mathbf{W}$, but $\mathcal{A}_{\mathbf{X},\mathbf{W},\mathbf{Y}}$ is marginally connecting, so there is a contradiction. $\square$

$\mathscr{R}_5$: If $\mathbf{X} *\!\!-\!\!* \mathbf{Z} *\!\!-\!\!* \mathbf{Y}$, $\mathbf{Z} - \mathbf{W}$, $\mathbf{X}$ and $\mathbf{W}$ are not adjacent, $\mathbf{Y}$ and $\mathbf{W}$ are not adjacent, and $\mathcal{A}_{\mathbf{X},\mathbf{Z},\mathbf{Y}}$ is never-connecting or conditionally-connecting with connection mark $\oplus_{\mathcal{D}}$ such that $\mathbf{W} \in \mathcal{D}$, then orient $\mathbf{Z} - \mathbf{W}$ as $\mathbf{Z} \to \mathbf{W}$.

*Proof.* The soundness of the rule comes from the definition of a connection mark, $\oplus_{\mathcal{D}}$, where any cluster $\mathbf{W} \in \mathcal{D}$ must be a descendant of a collider, such that $\mathbf{Z} \to \mathbf{W}$. $\square$

Next we prove orientation completeness for Rules 1-5.

**Lemma 3.** *Rules 1-5 collectively are complete in the sense that all orientations determined from successive application are valid and result in all possible orientations.*

*Proof.* In the case that there are no never-connecting arcs, by lemma 2 the rules are complete following Meek 1995 [9]. If there is one or more never-connecting arc, the orientation rules of CLOC result in fewer orientations, as never-connecting arcs always imply ambiguous orientations by remark 2. For any edge between $\mathbf{C}_i$ and $\mathbf{C}_j$ left undirected by successive applications of Rules 1-5, either the edge is part of a triplet marked with a marginally connecting or conditionally connecting arcs, or it is part of a triplet marked with a never-connecting arc. In the former case, by lemma 2 the cluster Markov equivalence class includes at least one model with $\mathbf{C}_i \rightarrow \mathbf{C}_j$ and at least one with $\mathbf{C}_i \leftarrow \mathbf{C}_j$. In the latter case, by remark 2, there exists at least one model in the cluster Markov equivalence class with $\mathbf{C}_i \rightarrow \mathbf{C}_j$ and at least one with $\mathbf{C}_i \leftarrow \mathbf{C}_j$.

Because CLOC and the orientation rules only make use of cluster level independence and dependence information, all marginal and conditional independencies for a given triplet are already evaluated. For a given triplet $\mathbf{C}_i, \mathbf{C}_k, \mathbf{C}_j$, by theorem 1, $\mathbf{C}_i$ and $\mathbf{C}_j$ can only be dependent if 1) they are adjacent, 2) they are not adjacent and $\mathcal{A}_{\mathbf{C}_i, \mathbf{C}_k, \mathbf{C}_j}$ is marginally connecting, 3) they are not adjacent, $\mathcal{A}_{\mathbf{C}_i, \mathbf{C}_k, \mathbf{C}_j}$ is conditionally connecting, and $\mathbf{C}_k$ is in the conditioning set, or 4) they are not adjacent, $\mathcal{A}_{\mathbf{C}_i, \mathbf{C}_k, \mathbf{C}_j}$ is never connecting, and there exists some descendant of a variable-level collider within $\mathbf{C}_k$ in cluster $\mathbf{C}_w$ where $\mathbf{C}_w$ is in the conditioning set. Cases 1, 2, and 3 are covered by the skeleton and collider search phases. Rule 5 captures conditional dependencies created by case 4, such that orientations for a non-oriented triplet can be made to reflect the dependence. As orientations of Rule 5 follow a non-standard pattern relative to Rules 1-3, we can consider information determined by Rule 5 to be a form of background knowledge introduced to the graph. Then, with Rule 4, and given the admissibility assumption of the partition, the proof for completeness extends directly from Meek 1995, where the PC algorithm with background knowledge is proved to be complete in that any subsequent orientations that can be determined following Rule 5 must be valid and complete. □

Finally, we prove Theorem 3 by showing that CLOC does return an $\alpha$C-CPDAG.

*Proof.* An $\alpha$C-CPDAG must reflect the cluster Markov equivalence class of $\alpha$C-DAGs for a given partition. This means that all cluster level independencies and dependencies must be represented, all directed edges are non-variant and all undirected edges are variant. The proof for non-variant directed edges and variant undirected edges follows from lemma 3. To represent all independencies and dependencies, we must ensure that all adjacencies, independence arcs, separation marks, and connection marks are determined. The proof for valid adjacencies follows directly from the proof for skeleton construction of Spirtes et. al 1993 [19]. The procedure for determining independence arcs follows from definition 2, where for each triplet, searches for variables in or not in the separating set for any given pair of variables $\mathbf{X}$ and $\mathbf{Y}$ allows for determination of the appropriate arc. The procedure for determining separation marks follows from definition 5, where independence tests are performed to identify where the closest pair of clusters, appearing to be dependent, are in fact independent. Lastly, the procedure for determining connection marks follows from definition 6, where independence tests are performed to determine if any combination of possible descendants render two variables dependents such that the set of clusters are necessarily descendants. Therefore, by theorem 2, the $\alpha$C-CPDAG completely represents a cluster Markov equivalence class. □

**Remark 3.** *CLOC is complete with background knowledge.*

*Proof.* The proof follows directly from the completeness of CLOC including the orientation rule (Rule 4) for background knowledge. □

## C   Further discussion on $\alpha$C-DAG semantics

### C.1   On separation marks, connection marks, and graph interpretation

In this section, we extend the discussion on the interpretation and semantics of $\alpha$C-DAGs.

We first further explore separation marks and connection marks. We note that separation marks can be placed on any independence arc that signifies a connection: marginally-connecting arcs, or

conditionally-connecting arcs. Separation marks can not be placed on never-connecting arcs, as there is no connection for the separation mark to dispute. When a separation mark is found on a marginally-connecting arc, a marginal connection is disputed. When a separation mark is on a conditionally-connecting arc, the connection, conditional on the center node of the triplet marked by the independence arc, is disputed. Since paths can be traversed in two directions, and independence statements can be read in two ways ($\mathbf{X} \perp\!\!\!\perp \mathbf{Y}, \mathbf{Y} \perp\!\!\!\perp \mathbf{X}$), separation marks come in pairs.

Connection marks are read in a way distinct from separation marks. The subscript of a connection mark indicates the directly connected nodes or sets of nodes that, when conditioned on, create a connecting triplet where there otherwise is not one. Any true descendants of the nodes in the subscript of the connection mark are understood to also create the connection, where a true descendant is identified by a true connecting path over clusters (see d-separation criteria, Def. 8). Connection marks can only be placed along never-connecting independence arcs. This is because a marginally active triplet can not have a new connection created due to conditioning on a descendant of a collider because the triplet is already active. If the center node of the triplet marked by a marginally-connecting independence arc is conditioned on, any descendant of a collider that is conditioned on would still fail to create a new connection as the independence arc necessitates there are non-colliders along any path the collider may appear on, which would be conditioned on, so the path would be blocked. As conditionally-connecting arcs require a collider, any true descendant will create a connection, following expected behavior, so there is no need to explicitly denote a connection mark. Lastly, we note that the subscript of a connection mark can be a set of sets of clusters. Each set of cluster denotes one way that the triplet can be made active, and it is noted that a path through a cluster with multiple colliders on it would need multiple descendants (possibly in different clusters) to be conditioned on for the triplet over clusters to be active.

There are certain graph semantics and attributes that require new interpretation for $\alpha$C-DAGs. In particular, we can create a more refined class of descendants and ancestors, informed by connections through the clusters. In C-DAGs, similarly as in DAGs and other graphs, a directed path from some node $\mathbf{C}_0$ to $\mathbf{C}_n$ is a sequence of distinct vertices $\langle \mathbf{C}_0, ..., \mathbf{C}_n \rangle$ such that for $0 \leq i \leq n-1$, $C_i$ is a parent of $C_{i+1}$ in $G_{\mathbf{C}}$. In $\alpha$C-DAGs, applying this same definition yields what we define as an **apparent directed path**, since even with the described pattern of edges, it is possible to have independence arcs and separation marks that describe a break or block which contradicts the notion of a directed path. By contrast a **true directed path** in an $\alpha$C-DAG from some node $\mathbf{C}_0$ to $\mathbf{C}_n$ is defined as a sequence of distinct vertices $\langle \mathbf{C}_0, ..., \mathbf{C}_n \rangle$ such that for $0 \leq i \leq n-1$, $\mathbf{C}_i$ is a parent of $\mathbf{C}_{i+1}$ in $G_{\mathbf{C}}$ and where every arc on the corresponding arc trajectory is a marginally-connecting arc with no separation marks. Then, $\mathbf{C}_A$ is called a **true ancestor** of $\mathbf{C}_B$ and $\mathbf{C}_B$ a **true descendant** of $\mathbf{C}_A$ if $\mathbf{C}_A = \mathbf{C}_B$ or there is a true directed path from $\mathbf{C}_A$ to $\mathbf{C}_B$. We contrast these terms with what we call **apparent ancestors** and **apparent descendants** where there may only be an apparent directed path from $\mathbf{C}_A$ to $\mathbf{C}_B$. In $\alpha$C-DAGs, we use the notation $An_{G_{\mathbf{C}}}(\mathbf{C}_B)$ and $De_{G_{\mathbf{C}}}(\mathbf{C}_A)$ to refer to the sets of **true ancestors** of $\mathbf{C}_B$ and **true descendants** of $\mathbf{C}_A$ in $G_{\mathbf{C}}$, respectively.

## C.2 On relaxing the assumption of acyclicity

In our definition of $\alpha$C-DAGs (and by extension for $\alpha$C-CPDAGs), we require that there is no **apparent cycle** over clusters, that is where for some pair of clusters $\mathbf{C}_i, \mathbf{C}_j$, where there exists an edge $\mathbf{C}_i \rightarrow \mathbf{C}_j$, there is no directed path $\mathbf{C}_j \rightarrow .... \rightarrow \mathbf{C}_i$. We believe this is a reasonable assumption in the context of clusters as the user intentionally defines the partition over variables, likely because these variables represent together some semantically meaningful entity or are otherwise similar in some ways, such that knowledge of a potential cycle is available. However, we also note that in some cases, such an assumption may not be feasible, and it is easy to construct an example where the underlying graph over variables is acyclic, but a certain partition over the variables creates an apparent cycle. In such a case, $\alpha$C-DAGs have the representational capacity to differentiate between a true cycle and an apparent cycle, as is clear by the discussion above differentiating between true and apparent ancestors and descendants. Specifically, if the assumption of acyclicity over clusters is relaxed (assuming an acyclic distribution over variables), then where there is an edge $\mathbf{C}_i \rightarrow \mathbf{C}_j$ and some directed path $\mathbf{C}_j \rightarrow .... \rightarrow \mathbf{C}_i$, there will necessarily exist some independence arc or separation mark along the path $\mathbf{C}_j \rightarrow .... \rightarrow \mathbf{C}_i$ that denotes that $\mathbf{C}_j$ is not a true ancestor of $\mathbf{C}_i$, and therefore there is no true cycle. In this context, properties such as d-separation extend soundly for $\alpha$C-DAGs. However, the relaxation of the assumption of no apparent cycles over clusters does have implications in the context of structure learning. In particular, rules that leverage this assumption of acyclicity are

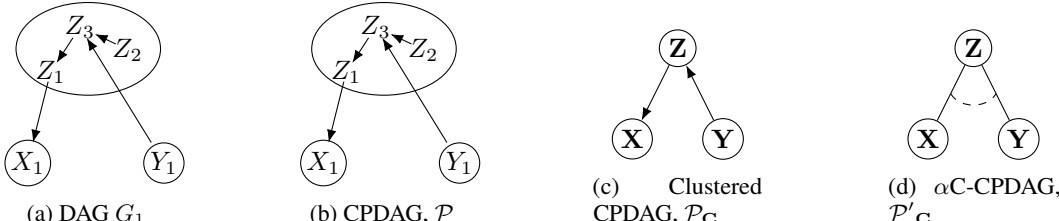

(a) DAG $G_1$  (b) CPDAG, $\mathcal{P}$  (c) Clustered CPDAG, $\mathcal{P}_{\mathbf{C}}$  (d) $\alpha$C-CPDAG, $\mathcal{P}'_{\mathbf{C}}$

Figure 6: $(a)$ is a DAG and $(b)$ is the CPDAG that comes from $G_1$. Following the procedure in definition 13, $(c)$ is the clustered CPDAG that comes from $\mathcal{P}$. This object reflects orientations that are determined from tests on $P(\mathbf{V})$. By contrast, $(d)$ is the $\alpha$C-CPDAG that corresponds to $G_1$. All edges are undirected as $\mathbf{X} \not\perp \mathbf{Y}; \mathbf{X} \perp\!\!\!\perp \mathbf{Y}|\mathbf{Z}$ and the edges cannot be oriented as, by Remark 1, the cluster level dependencies and independencies align with the representations of $\mathbf{X} \to \mathbf{Z} \to \mathbf{Y}$, $\mathbf{X} \leftarrow \mathbf{Z} \to \mathbf{Y}$, and $\mathbf{X} \leftarrow \mathbf{Z} \leftarrow \mathbf{Y}$.

no longer valid, such as Rule 2 and Rule 4. Rule 3 depends upon the validity of Rule 2 and therefore also becomes invalid. An area of future work is to determine sound extension of or different rules that allow for sound and complete learning over clusters when the acyclicity assumption is relaxed. In Appendix D we show analysis on the number of wrongly-oriented edges when CLOC is run on a cyclic partition.

### C.3 On the special case of clusters of size 1

We note that when all clusters include at most 1 variable, CLOC reduces to PC, following Lemma 2. Independence arcs, separation marks, and connection marks all become redundant. When clusters have more than 1 variable, and there are no never-connecting arcs, the orientation rules also reduce to PC, however the graphical object still requires separation and connection marks to fully represent conditional independences and dependences. When clusters have at most 1 variable, this is no longer the case. For any triplet $\langle \mathbf{C}_i, \mathbf{C}_k, \mathbf{C}_j \rangle$ such that $\mathbf{C}_k$ is of size $n = 1$ (i.e. there is only one variable in the cluster), the alignment of the edge orientations and marginal and conditional independences and dependences will be aligned as the case is for variables. For a simplified representation in $\alpha$C-DAGs and $\alpha$C-CPDAG, independence arcs and connection marks could be removed for these triplets. The interpretation of this object is that wherever there is an omitted independence arc, the behavior for the triplet is as anticipated. If there exists another triplet in the graph $\langle \mathbf{C}_r, \mathbf{C}_q, \mathbf{C}_w \rangle$ such that $\mathbf{C}_q$ is not of size $n = 1$, it is possible a separation mark is required for $\langle \mathbf{C}_i, \mathbf{C}_k, \mathbf{C}_j \rangle$, in which case the independence arc, with the appropriate separation mark, would be required. If all clusters in an $\alpha$C-DAG or $\alpha$C-CPDAG include at most 1 variable, then the simplified representation holds for all triplets and the result would be a DAG or CPDAG respectively.

## D Experimental details and additional results

### D.1 Experimental Setup

All experiments were run on a machine with CPU: Intel i9 Chip, 32 GB of RAM, and macOS operating system. A single core was used for the experiments. Algorithms are implemented in Python and implementation of CLOC and experiments are available at: https://github.com/TaraAnand/CLOC

In our simulations, we compare two approaches to developing a clustered graphical equivalence class. The first approach consists of applying PC to the distribution over variables, $P(\mathbf{V})$, and then imposing clusters. The clustering procedure is shown below.

**Definition 13** (**Clustered CPDAG.**). *Given a CPDAG, $\mathcal{P}$ over variables $\mathbf{V}$, and a partition $\mathbf{C} = \{\mathbf{C}_1, ..., \mathbf{C}_n\}$ of $\mathbf{V}$, construct a graph $\mathcal{P}_{\mathbf{C}}$ over $\mathbf{C}$ as follows.*

- *An edge $\mathbf{C}_i \to \mathbf{C}_j$ is in $\mathcal{P}_{\mathbf{C}}$ if there exists some $V_i \in \mathbf{C}_i$ and some $V_j \in \mathbf{C}_j$ such that $V_i \in Pa(V_j)$ in $\mathcal{P}$*
- *An edge $\mathbf{C}_i - \mathbf{C}_j$ is in $\mathcal{P}_{\mathbf{C}}$ if for all $V_i \in \mathbf{C}_i$ that are adjacent to some $V_j \in \mathbf{C}_j$, there is an undirected edge between $V_i$ and $V_j$, i.e. $V_i - V_j$.*

We note that the graphical object created by the procedure above, which we refer to as a clustered CPDAG, determined by the PC-then-Cluster approach, is distinct from an $\alpha$C-CPDAG. In particular,

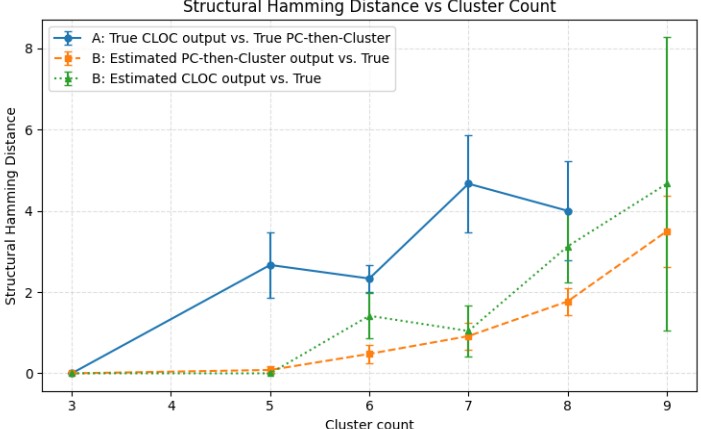

Figure 7: Green: comparison of CLOC output, estimated from a simulated Gaussian dataset, compared to the oracle for the corresponding data-generating process. Orange: comparison of the PC-then-Cluster approach output, estimated from a simulated Gaussian dataset, compared to the oracle for the corresponding data-generating process. Blue: Comparison of the oracle solutions by CLOC and the PC-then-Cluster approach.

edges that may in fact be variant in a cluster Markov equivalence class may become oriented in the clustered CPDAG, due to some feature of the distribution over variables. For example, in Figure 6, the distribution over variables, $P(\mathbf{V})$ allows the collider over $\langle Z_2, Z_3, Y_1 \rangle$ to be learned, allowing for an orientation between $\mathbf{Y}$ and $\mathbf{Z}$ to be possible for the clustered CPDAG. Subsequent applications of Rule 1 of the PC algorithm allows for orientation of the edge $Z_1 \rightarrow X_1$, so that an orientation between $\mathbf{X}$ and $\mathbf{Z}$ is possible. By contrast, the $\alpha$C-CPDAG is learned from the distribution $P(\mathbf{C})$ where cluster-level independence tests reveal $\mathbf{X} \not\perp\!\!\!\perp \mathbf{Y}; \mathbf{X} \perp\!\!\!\perp \mathbf{Y}|\mathbf{Z}$. The cluster Markov equivalence class for this information includes graphs with the orientations $\mathbf{X} \rightarrow \mathbf{Z} \rightarrow \mathbf{Y}$, $\mathbf{X} \leftarrow \mathbf{Z} \rightarrow \mathbf{Y}$, and $\mathbf{X} \leftarrow \mathbf{Z} \leftarrow \mathbf{Y}$, so no orientations in the $\alpha$C-CPDAG can be made.

For the experiments in the main body of the paper, we compare the methods of CLOC and the PC-cluster approach, as there is no other comparable method outputting an equivalence class over clusters. For the latter method, we use the built-in implementation of PC in the python package causal-learn [27]. The output is a CPDAG, which is then clustered by the procedure described in definition 13 using the defined partition over variables into clusters. In our implementation of CLOC the multi-variate conditional independence test used iterates over pair-wise tests of variable level independence tests with early stopping when a dependence is determined implying dependence over clusters.

## D.2 Additional results

We show additional experimental results in Figure 7. In comparing oracle (ground truth) results by the PC-then-cluster approach with CLOC, we can note information that is lost by using only cluster-level information rather than variable-level information. As is illustrated in Figure 6, orientations beyond those representing the cluster Markov equivalence class are possible when the (variable-level) Markov equivalence class is learned by leveraging $P(\mathbf{V})$. In Figure 7 The blue line shows how much of this sort of information, translating to orientations aligning with $P(\mathbf{V})$, is lost when only $P(\mathbf{C})$ is used. We expect this number to be non-zero. This tradeoff in orientation capacity can be weighed against improvements in required number of conditional independence tests and runtime, as demonstrated in the main body.

The green and orange lines compare, for each method of CLOC and the PC-then-cluster approach, the structural hamming distance between a graph estimated from a data sample as compared to the ground truth equivalence class. We note that we see similar structural hamming distances for CLOC compared to the PC-then-Cluster approach, which reflects similar robustness of our proposed method to noise in data samples, despite larger conditioning sets.

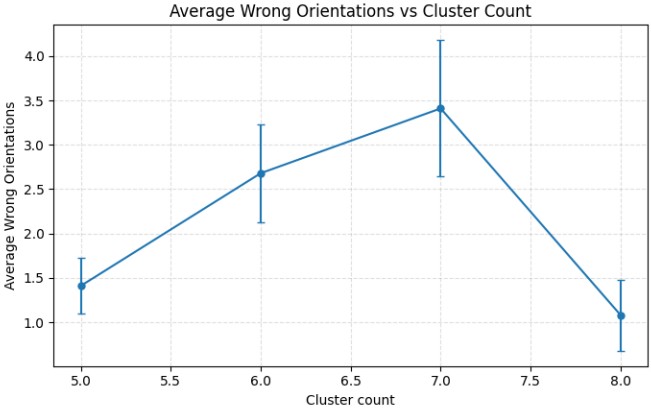

Figure 8: Average number of misoriented edges for graphs with 5, 6, 7, and 8 clusters with inadmissible (cyclic) partitions across 100 runs.

We also run analyses to show the impact of violations to the assumption about an acyclic partition. Specifically, for random graphs with acyclic partitions, corresponding to $\alpha$C-DAGs, we generate a different partition over the graph that induces a cycle. Using the parent-child relationships of the true DAG, we assess how many wrong orientations (where an edge that is oriented contradicts the true direction) are determined by running CLOC with the inadmissible partition. We show results averaged over 100 simulations each for cyclic partitions of 5, 6, 7, and 8 clusters. The results are shown in Figure 8.

## E  Complexity Analysis

The skeleton construction requires, for each pair of clusters $(\mathbf{X}, \mathbf{Y})$, searching for conditional independence given subsets of their neighboring clusters. The number of possible conditioning sets grows combinatorially with degree $d$, the maximum degree of any cluster, so there are $O(2^d)$ possible subsets to check per pair. For a graph with $n$ clusters and $\binom{n}{2} = O(n^2)$ pairs, there are a total of $O(n^2 2^d)$ tests. Independence arcs are then determined for each triplet $\langle \mathbf{X}, \mathbf{Z}, \mathbf{Y} \rangle$. The number of unshielded triplets for which an additional test is needed is bounded by $O(nd^2)$. The number of shielded triplets for which an additional test is needed is bounded by $O(nd^3)$, as four nodes are involved in these tests. The search for separation marks, we should note, is not necessary for determining graphical orientations. However, to create a complete $\alpha$C-CPDAG on which subsequent analyses can be done, separation marks are necessary. Assuming a longest path length $L$, the search is bounded by $O(nd^L)$. Connection marks' search can also be expensive and the marks are informative for edge orientations. In practice, the size of the conditioning set $\mathbf{W}$ can be bounded to save costs. In the worst case, each triplet is evaluated for all subsets of neighbors of $\mathbf{Z}$, yielding $O(nd^2 2^d)$. The last algorithmic component of evaluating the orientation rules until none apply requires searching over all triplets, bounded by $O(nd^2)$. In total, the complexity is bounded by $O(n2^d(n + d^2))$. Where clusters are created such that inter-cluster density is relatively sparse and intra-cluster density is high, CLOC will show the greatest complexity benefits relative to PC.

