# OpenReview forum: "Causal Discovery over Clusters of Variables in Markovian Systems"
_NeurIPS.cc/2025/Conference — NeurIPS 2025 poster_

### Official Review · Reviewer_2UQT · 2025-06-23

**Clarity:** 3
**Significance:** 3
**Originality:** 2
**Rating:** 4
**Confidence:** 3

**Summary:**

This paper introduces a new graphical model, the αC-DAG, designed to learn causal relationships between predefined clusters of variables from observational data. It incorporates independence arcs, separation marks, and connection marks to represent conditional independence patterns among clusters and defines a custom d-separation criterion. The authors also propose αC-CPDAGs to capture equivalence classes of these graphs, along with a corresponding learning algorithm called CLOC. They prove the soundness of the framework and show that CLOC performs more efficiently than a PC-then-cluster baseline on synthetic data.

**Questions:**

1. How sensitive is the method to small errors or noise in the cluster partition? Would a few misclassified variables significantly distort the learned structure?
2. Can the approach be extended to support overlapping or hierarchical clusters, which are common in domains like text modeling?
3. In cases where conditional independence tests produce conflicting results (due to statistical noise or small samples), how are contradictory arc annotations resolved?
4. Does the graphical formalism still maintain interpretability and scalability when applied to systems with dozens or hundreds of clusters?

**Ethical Concerns:**

["NO or VERY MINOR ethics concerns only"]

**Final Justification:**

This paper introduces a new graphical model, the αC-DAG, designed to learn causal relationships between predefined clusters of variables from observational data. Although the response may not address the concerns about the strong assumptions relied upon and the difficulty of constraining real-world systems, it did demonstrate the plausibility of the current assumptions. Therefore, I decided to slightly increase my score.

**Limitations:**

1. The approach assumes causal sufficiency at the cluster level. These assumptions are strong and often unrealistic in practical settings, particularly when latent confounding or weak interactions exist across clusters.
2. The algorithm requires the cluster partition to be given and acyclic, which may not always be feasible. No method is proposed to validate, refine, or learn the partition itself.
3. The symbolic structure of αC-DAGs (e.g., independence arcs, separation marks, connection marks) adds considerable complexity. The usability of such graphs in practical decision-making or downstream tasks like intervention planning is not demonstrated.

**Paper Formatting Concerns:**

There are no paper formatting concerns in this paper.

**Quality:**

2

**Strengths And Weaknesses:**

Strengths
1. The paper introduces a novel graphical model, the αC-DAG, and its equivalence class representation, the αC-CPDAG, specifically designed to capture causal relationships between predefined clusters of variables. The framework adds expressive symbolic elements—independence arcs, separation marks, and connection marks—that allow for a more nuanced encoding of conditional independencies at the cluster level.
2. The authors establish sound and complete d-separation semantics for αC-DAGs, define a precise notion of cluster-level Markov equivalence, and prove the correctness of their orientation rules and learning algorithm.
4. The experiments demonstrate meaningful comparisons against a standard PC-then-cluster baseline. Results show that the proposed approach achieves comparable accuracy.


Weaknesses
1. The method assumes a provided clustering of variables that is both acyclic and accurate. This assumption directly encodes structural information, which contradicts the premise of discovering causal structure from observational data. As a result, the learning process becomes more of a structural summarization under a fixed partition than genuine discovery.
2. The graphical formalism of αC-DAG blends causal relationships and statistical independencies into a single representation. While this increases expressive power, it also blurs the distinction between mechanistic causality and observed conditional dependence, potentially complicating the interpretation and soundness of orientation rules derived from such hybrid semantics.
3. The model relies heavily on conditional independence tests to annotate arcs and guide structure learning. However, multivariate CI tests are known to be statistically unstable, especially with small samples or non-Gaussian distributions, and the framework does not incorporate any robustness measures or uncertainty modeling.

---

> ### Author Rebuttal · Authors · 2025-07-30
>
> We thank the reviewer for their time in reading our paper. We believe a few misunderstandings may have contributed to a somewhat harsh evaluation, and kindly ask the reviewer to reconsider our work in light of the clarifications below.
>
> **W1**
> >The method assumes a…clustering…that is both acyclic and accurate. This assumption directly encodes structural information, which contradicts the premise of discovering causal structure from observational data. As a result, the learning process becomes more of a structural summarization under a fixed partition than genuine discovery.
>
> While CLOC relies on a given partition, this form of limited background knowledge is integrated with data-driven learning. The partition informs structure, but is weaker knowledge than even ordering; performing discovery is still required. For concreteness, consider a simple graph of 3 clusters where the true graph is an unshielded triplet. In this case, knowledge of acyclicity provides no useful information. The collider test remains necessary; if the structure is not a collider, the partition doesn’t indicate if the triplet forms a direct chain, a reverse chain, or a fork.
>
> **W2**
> >The graphical formalism of αC-DAG blends causal relationships and statistical independencies into a single representation. While this increases expressive power, it also blurs the distinction between mechanistic causality and observed conditional dependence, potentially complicating the interpretation and soundness of orientation rules derived from such hybrid semantics.
>
> Independence arcs in αC-DAGs encode conditional independencies, which come from knowledge. In αC-CPDAGs, arcs (and edges) are learned from data. The goal of causal discovery is to leverage independencies from data and graphical properties to establish causal relationships. In other causal graphs (DAGs, CPDAGs, ADMGs, PAGs), edge orientations encode both causality and independencies, as these are intrinsically linked. This key observation from Spirtes, Glymour, Schines, and Pearl started the literature of structure learning. Therefore in our cluster context, an accurate representation of independence is critical, and one of the key results established in our paper with independence arcs and separation/connection marks. We prove soundness of our orientation rules and show the logical alignment of our semantics with the standard graphical properties on which the orientation rules depend.
>
> **W3**
> >The model relies heavily on conditional independence tests to annotate arcs and guide structure learning. However, multivariate CI tests are known to be statistically unstable, especially with small samples or non-Gaussian distributions, and the framework does not incorporate any robustness measures or uncertainty modeling.
> Causal discovery remains a challenging task, primarily due to high computational complexity and reliance on the accuracy of statistical tests. As the number of variables increases, causal discovery at the variable-level quickly becomes infeasible, even with large datasets. Our work shows that this computational burden can be alleviated by discovery over clusters, which reduces the dimensionality of the problem. This shift allows focus on the challenge of accurately inferring statistical associations, a task that becomes more tractable with larger sample sizes and modern statistical/machine learning techniques. We also show that our method maintains robustness compared to methods leveraging tests over variables alone (Figure 7, Appendix C.2) and emphasize that this is the first algorithm for learning C-DAGs. As in the structure learning literature, leveraging imperfect statistical tests often enables tractable solutions to foundational problems. Over time, these methods are improved, but progress begins with principled starting points. We appreciate the reviewer’s concern, and agree that developing stable multivariate CI tests is a compelling open challenge, which we hope this work helps motivate.
> **Q1**
> >How sensitive is the method to small errors or noise in the cluster partition? Would a few misclassified variables significantly distort the learned structure?
>
> Partitions can be created however a user wishes, as long as a cycle is not induced. Therefore, it is difficult to say what would constitute errors or noise in a partition, except to say that a certain partition induces a cycle. CLOC is sensitive to cyclic partitions, as our assumptions state. We conducted additional experiments illustrating the extent of this sensitivity (see table in response to RBGk, omitted here due to space constraints).
>
> We believe there are meaningful real-world settings where knowledge of an admissible partition is available. For instance, in medicine, partitions often arise naturally: clusters might correspond to laboratory panels (e.g. blood tests), neuroanatomical regions, or demographic blocks. Temporal or biological ordering provides good reason to assume acyclicity over groups.
>
> **Q2**
> >Can the approach be extended to support overlapping or hierarchical clusters, which are common in domains like text modeling?
>
> We have not yet explored overlapping clusters, as the nature of the solution changes substantially in this context. However, overlapping variables is an interesting and practical challenge that we hope to see further explored. The question/suggestion is appreciated.
>
> **Q3**
> >In cases where conditional independence tests produce conflicting results (due to statistical noise or small samples), how are contradictory arc annotations resolved?’
>
> It should be noted that algorithms like PC, on which our approach is based, do not note or aim to resolve conflicts that might appear. Each conditional independence test is assumed correct, and propagates; tests that might contradict earlier results are never conducted. Further in CLOC, each independence arc pertains to the triplet of clusters that it annotates, and a single pair of independence tests determines the arc, such that there could be no conflicts. For edges and separation/connection marks with possibly contradicting evidence in the data, CLOC proceeds as standard PC, assuming all previous tests are correct, so there are no conflicts to resolve although there could be errors.
>
> **Q4**
> >Does the graphical formalism still maintain interpretability and scalability when applied to systems with dozens or hundreds of clusters?
>
> There is nothing in principle that prevents the application of our proposed graphical formalism and algorithm to systems with dozens or hundreds of clusters. While interpretability is often a matter of design choices and domain expertise, scalability is not inherently guaranteed. Learning a causal graph is a super-exponential problem in the number of nodes. Thus, as the number of cluster-nodes increases, the computational complexity can grow rapidly. Although clustering helps reduce dimensionality compared to working with individual variables, learning a graph over many clusters can still entail significant computational and statistical challenges.
>
> **L1**
> >The approach assumes causal sufficiency at the cluster level. These assumptions are strong and often unrealistic in practical settings, particularly when latent confounding or weak interactions exist across clusters.”
>
> We acknowledge that causal sufficiency is a strong assumption. Solving this first step has laid a strong foundation for addressing problems where latent confounding exists. Having said that, the latent confounding case is much more involved and substantially harder to discuss without a thorough understanding of the Markovian case.
>
> **L2**
>  > The algorithm requires the cluster partition to be given and acyclic, which may not always be feasible. No method is proposed to validate, refine, or learn the partition itself.
>
> Learning admissible partitions is certainly a task of interest to us, but it is beyond the scope of this work. A given acyclic partition can be considered a specific form of background knowledge that is integrated into the learning process. As discussed in the response to Q1, there are many use cases where such knowledge is reasonable given the domain.
>
> **L3**
> > The symbolic structure of αC-DAGs (e.g., independence arcs, separation marks, connection marks) adds considerable complexity. The usability of such graphs in practical decision-making or downstream tasks like intervention planning is not demonstrated.
>
> We agree that the symbolic structure of αC-DAGs introduces additional notation and complexity. However, this is a deliberate design choice motivated by scalability and interpretability. In high-dimensional domains such as drug safety, social science, or genomics, conventional DAGs or PAGs are often impractical to elicit, interpret, or reason over.
> By introducing coarse-grained structures (grouping variables into meaningful clusters such as drug classes, social determinants, or anatomical regions), αC-DAGs dramatically reduce the effective dimensionality. While the use of symbolic marks adds some complexity, it enables a compact and expressive representation of high-level causal relationships across clusters. This can make the analysis of such systems tractable where it would otherwise be infeasible.
> Additionally, αC-DAGs need not be directly parsed by humans for all use cases. They can serve as intermediate representations for downstream tasks like interventional or counterfactual inference and can support quantitative reasoning in automated systems.
> αC-DAGs introduce a new point in the tradeoff between expressiveness, interpretability, and scalability. In cases where variable-level DAGs are impractical to learn or use, structured coarse representations may provide a valuable and feasible alternative.

---

> > ### Comment · Reviewer_2UQT · 2025-08-05
> >
> > Thank you for your response, which has addressed some of my concerns. Therefore, I have decided to raise my score. However, considering the strong assumptions underlying this paper, I suggest that the authors further discuss the constraints in real-world systems to enhance the applicability of the work.

---

### Official Review · Reviewer_rfuq · 2025-06-23

**Clarity:** 2
**Significance:** 4
**Originality:** 4
**Rating:** 5
**Confidence:** 4

**Summary:**

The paper tackles the problem of learning Cluster-DAGs (C-DAGs) from observational data, where a C-DAG is defined as the clustering of a low-level probabilistic causal model. The authors highlight how the connection between $d$-separations and conditional independence relations does not typically hold when considering clusters of variables. To this end, they introduce the novel notion of *independence arcs*, i.e., an arc connecting two edges of a C-DAG. The authors show how conditional independences between clusters can be expressed as a function of paths composed of independence arcs. Using this result, they design a constraint-based method — named CLOC — to recover C-DAGs from data, up to an equivalence class analogous to the Markov Equivalence Class for DAGs.

**Questions:**

The scope and the contributions of the work are sufficiently clear, and I do not have particular questions for the authors.

**Ethical Concerns:**

["NO or VERY MINOR ethics concerns only"]

**Final Justification:**

The authors sufficiently discussed the identified weaknesses of their paper, which constituted minor points in an overall positive evaluation of the work.

**Limitations:**

yes

**Paper Formatting Concerns:**

None.

**Quality:**

3

**Strengths And Weaknesses:**

Strengths:
- The paper well-motivates the introduction of their method, showing how naively applying causal discovery algorithms, such as PC, might not return the ground-truth C-DAG.
- The introduction of independence arcs is interesting, novel, and correctly captures conditional independences among clusters.
- Under the commonly adopted faithfulness assumption, the CLOC algorithm provably returns the equivalence class of the ground-truth C-DAG with the corresponding independence arcs, i.e., an $\alpha$ CP-DAG in the terminology of the authors.

Weaknesses:
- The work shares the limitation of existing works on C-DAGs, by assuming that the clusters are provided. A practitioner does not know how variables are causally related, but must know how to cluster them.
- The notation of independence arcs is quite confusing, and figures such as Figure 3 are difficult to interpret. For instance, overlapping symbols in Figure 3d — see for instance between clusters $\mathbf{W}, \mathbf{Y}$ — make it quite unreadable.

---

> ### Author Rebuttal · Authors · 2025-07-30
>
> Thank you for reading our paper and acknowledging the significance and contributions of our work. We appreciate the reviewer feedback and provide a few notes below.
>
> ***W1:***
> >The work shares the limitation of existing works on C-DAGs, by assuming that the clusters are provided. A practitioner does not know how variables are causally related, but must know how to cluster them.
>
> It is true that, like with existing work on C-DAGs, our algorithm requires a pre-specified partition of variables. C-DAGs, where the context is identification and inference, provide an alternative to constructing a complete causal diagram, and the assumption of a known partition over variables is a less stringent requirement than knowing the full causal diagram over all variables (while there are tradeoffs). In our work, with a focus on the task of learning, we believe the assumption of a known partition can offer improvements in scalability such that knowledge of the partition is a meaningful tradeoff. It is true that the approach may not be applicable in all cases if knowledge of a partition proves infeasible. However, we believe this requirement is both meaningful and practical in a range of real-world settings. For instance, in clinical and biological domains, partitions often arise naturally: clusters might correspond to laboratory panels (e.g., blood tests), gene sets, microbiome groups, neuroanatomical regions, or demographic blocks. In many of these domains, temporal or biological ordering provides good reason to assume acyclicity at the group level.
>
> In other words, C-DAGs offer a relaxation over DAGs, albeit not universally applicable to all modeling scenarios, and the current work on learning presents a possible relaxation/different type of tradeoff compared to other competing methods. Having said that, we agree that the proposed method does not constitute a universal solution to all learning problems in causality.
>
> **W2:**
> > The notation of independence arcs is quite confusing, and figures such as Figure 3 are difficult to interpret. For instance, overlapping symbols in Figure 3d — see for instance between clusters \mathbf{W}, \mathbf{Y} — make it quite unreadable.
>
> We acknowledge that the independence arcs can result in dense graph visuals and apologize for having to use such notation. Having said that, we note that this notation is made out of necessity, since familiar graph edges are incapable of representing both ancestral relationships between clusters and the conditional independence information between, and both of these types of information are important for structure learning and downstream tasks. This realization and the operationalization of the additional kind of arcs constitute a key contribution of this work. Still, we will revise the discussion and figures to improve the readability of the notation and make this point clearer; thank you.

---

> > ### Comment · Reviewer_rfuq · 2025-08-03
> >
> > I thank the authors for their discussion on C-DAGs and their choice of improving readability of figures. Overall, I maintain my positive evaluation of their work.

---

### Official Review · Reviewer_RGBk · 2025-07-01

**Clarity:** 2
**Significance:** 3
**Originality:** 3
**Rating:** 5
**Confidence:** 2

**Summary:**

The authors introduce αC-DAGs, a novel graphical representation that augments cluster DAGs (C-DAGs) with three annotations: independence arcs, separation marks, and connection marks. These additions allow αC-DAGs to fully encode conditional independence relationships over clusters, which is not possible with standard C-DAGs. The paper establishes soundness and completeness of a new d-separation criterion for αC-DAGs (Theorem 1). It then defines αC-CPDAGs as a representation for Markov equivalence classes of αC-DAGs. The CLOC algorithm is proposed for learning αC-CPDAGs from observational data, leveraging constraint-based methods and five orientation rules (R0-R5), which are proven sound. Experiments on synthetic data (up to 256 variables and 8 clusters) show that CLOC matches the cluster-level accuracy of the standard PC algorithm applied to the full variable set and then aggregated, while requiring fewer conditional independence tests and less computational time.

**Questions:**

-Why are clusters of nodes ultimately integrated into a graph over variables representing a class of DAGs? Why is there no other structure or cycle?

-How sensitive is CLOC to misspecified partitions? You should have done an obligation study.

-How did you perform multivariate CI tests?

**Ethical Concerns:**

["NO or VERY MINOR ethics concerns only"]

**Final Justification:**

Thank you, authors, for the rebuttal. It answered most of my concerns. However, the manuscript should be updated as promised.

**Limitations:**

As the authors mentioned, users are required to have knowledge of a partition of variables that does not induce a cycle, which is very limiting in practice.

There is an experimental or theoretical analysis of how high-dimensional CI tests impact CLOC performance.

**Paper Formatting Concerns:**

-Experiment figures are difficult to read at 100 % zoom; consider larger fonts for labels and numbers.
No legend for the plots!

-Line 65 : slearn -> learn

-Be consistent with the definitions in the preliminary. Why is the keyword "Definition." in the middle of line 80, while it was not used before? You should either make a new paragraph or be consistent.

**Quality:**

3

**Strengths And Weaknesses:**

The αC-DAG representation and the associated d-separation criterion are, to the best of my knowledge, new contributions. The paper has formal definitions and theorems (e.g., soundness and completeness of d-separation in αC-DAGs) and proofs (which I have not read). They also provided the CLOC algorithm, which is well-specified, and the orientation rules are motivated and proven sound.

Weakness:

The approach relies on user-provided clusters or partitions that must form a DAG. The paper does not address how such a partition is obtained in practice. If the partition is misspecified (e.g., clusters are not acyclic), the method fails, right? This limits practicality, especially since cluster-level relationships might be unknown a priori, and the user has no idea if there is a cycle or not.

Theorem 3 shows soundness but leaves open whether the αC-CPDAG always captures the full equivalence class. This is a critical gap because without completeness, the learned graph might be less informative than possible. Right?

Experimental comparison overlooks recent methods for cluster-level causal discovery techniques; why is no other comparison provided?
Also, there is no real-world validation. I also expected to see an obligation study on misspecified clusters or partitions.

The presentation was a bit hard to follow for me with many notations and definitions. I am not sure if you can relax some notations to improve readability.

---

> ### Author Rebuttal · Authors · 2025-07-30
>
> Thank you for taking the time to read our paper and for acknowledging the novelty and soundness of our contributions. We comment on the concerns raised in the sequel.
>
> ***W1:***
> > The approach relies on user-provided clusters or partitions that must form a DAG. The paper does not address how such a partition is obtained in practice. If the partition is misspecified (e.g., clusters are not acyclic), the method fails, right? This limits practicality, especially since cluster-level relationships might be unknown a priori, and the user has no idea if there is a cycle or not.
>
> Thank you for raising this important point. Our method indeed assumes a user-provided, admissible partition over variables, and if the partition is misspecified (e.g., the cluster graph contains cycles), the method may fail to recover a correct C-DAG. Two points are worth making in this context.
> First, we believe this requirement is both meaningful and practical in a range of real-world settings. For instance, in clinical and biological domains, partitions often arise naturally: clusters might correspond to laboratory panels (e.g., blood tests), gene sets, microbiome groups, neuroanatomical regions, or demographic blocks. In many of these domains, temporal or biological ordering provides good reason to assume acyclicity at the group level. Our framework is designed to formalize, allow for, and exploit such domain knowledge whenever available.
> Second, and more broadly, we emphasize that this is the first algorithm for learning C-DAGs -- a class of models that combines coarse-grained structure with interpretability and modularity. As with the early days of PC/FCI, and in the structure learning literature, strong assumptions often enable tractable solutions to foundational problems. Over time, these assumptions can be relaxed, and robustness added, but progress begins with principled starting points. We are still trying to improve these algorithms more than three decades after they were proposed.
> We appreciate the reviewer’s concern and agree that developing methods for verifying or learning the cluster-level structure is a compelling open challenge. We hope this work serves as a foundation for that line of research.
>
> ***W2:***
> >Theorem 3 shows soundness but leaves open whether the αC-CPDAG always captures the full equivalence class. This is a critical gap because without completeness, the learned graph might be less informative than possible. Right?
>
> We agree that completeness is important for ensuring the most informative learned graph. After submitting our main manuscript, we did manage to prove CLOC’s completeness, as detailed in Lemma 2 and Remark 5 in Appendix A. We will update the theorem description in the main body of our paper accordingly.
>
> ***W3:***
> >Experimental comparison overlooks recent methods for cluster-level causal discovery techniques; why is no other comparison provided? Also, there is no real-world validation. I also expected to see an obligation study on misspecified clusters or partitions.
>
> To our knowledge, there are currently no causal discovery techniques that learn a graphical object over clusters. Such a graphical equivalence class has not been previously defined, and is one of the major contributions of this work. As discussed in Section 1.1, related work is limited to methods that either use clusters as an intermediate step for variable-level causal discovery or target narrow scenarios, such as those with only two clusters. These approaches are not designed for the general, cluster-level causal discovery setting that CLOC addresses and therefore do not offer a meaningful basis for comparison. Other related works have simply identified the challenges of learning over clusters, without proposing an algorithm to address these challenges as our work does. We agree that conducting a sensitivity analysis on misspecified partitions would be valuable and have done so. We include more details in response to Q2.
>
> ***W4:***
> >The presentation was a bit hard to follow for me with many notations and definitions. I am not sure if you can relax some notations to improve readability.
>
> We apologize if parts of the paper were difficult to follow – that was certainly not our intention. We aimed to keep the formalization as concise as possible while introducing the key components needed to define the framework and support our main results (e.g., different levels of coarsening over the original causal DAG, the appropriate equivalence class, a modified notion of separation, faithfulness, proper invariances, etc). We appreciate the reviewer’s comment and understanding, and will make every effort to improve the clarity and readability of the notation, should the paper be accepted and an additional page be made available (as in previous years).
>
> ***Q1:***
>  >Why are clusters of nodes ultimately integrated into a graph over variables representing a class of DAGs? Why is there no other structure or cycle?
>
> We agree that alternative representations, potentially involving cycles, could offer greater flexibility and support arbitrary clusterings. However, this setting introduces significant challenges that require extensive learning over individual variables, thereby counteracting the computational and robustness advantages of cluster-level discovery. Enforcing acyclicity at the cluster level is a mild yet effective assumption that ensures scalability and interpretability without reintroducing the complexity of variable-level learning. Having said that, we think there is space for broader exploration and we take the reviewer’s note as an intriguing suggestion.
>
> ***Q2:***
> >How sensitive is CLOC to misspecified partitions? You should have done an obligation study.
>
> CLOC can be sensitive to misspecified partitions that induce a cycle. Aside from this requirement, variables can be partitioned in any way. We have conducted additional experiments illustrating the extent of the sensitivity to inadmissible partitions. We include some results in the table below, which will be included in the updated manuscript. In words, the table shows, for different numbers of clusters in the graph, the impact of specifying the partition such that a cycle is induced. We evaluate this by the structural hamming distance (SHD) of the graph determined by the oracle using the true partition compared to the graph determined by the oracle using the misspecified partition, and report an average over 200 iterations of random graphs. We compare this to the SHD of an oracle graph of the initial partition specified to a context where a variable is arbitrarily moved from one cluster to another, to illustrate the relative degree to which a change is expected. As discussed in the response to W1, the goal is to integrate prior knowledge in the form of an admissible partition; there are many use cases where such prior knowledge is feasible, and these are the intended use cases of CLOC.
>
> | # Clusters      | # Nodes | Avg SHD (SD) - Cyclic     | Avg SHD (SD) - Acyclic     |
> | :---:        |    :----:   |          :---: |           :---: |
> | 5 | 32 | 1.495 (1.248) | 1.240 (1.183) |
> | 6 | 64| 1.720 (1.442) | 1.930 (1.454)  |
> | 7 | 128	| 3.100 (1.857) | 2.800 (1.779) |
> | 8 | 256	| 3.570 (2.354) | 3.265 (2.430)|
>
>
> ***Q3:***
> >How did you perform multivariate CI tests?
>
> Multivariate CI tests were performed using an approach of iteration through pairwise correlations between variables with early stopping when a dependence is found, an approach appropriate for the Gaussian distribution. This is discussed in Appendix C.1.
>
> ***PF:***
> >Experiment figures are difficult to read at 100\% zoom; consider larger fonts for labels and numbers. No legend for the plots! Line 65 : slearn -> learn. Be consistent with the definitions in the preliminary. Why is the keyword "Definition." in the middle of line 80, while it was not used before? You should either make a new paragraph or be consistent.
>
> Thank you for these notes. We will make the suggested formatting edits.

---

### Official Review · Reviewer_GBnh · 2025-07-02

**Clarity:** 3
**Significance:** 3
**Originality:** 3
**Rating:** 4
**Confidence:** 4

**Summary:**

This paper addresses the challenge of causal discovery in high-dimensional settings by proposing an approach based on clustered variables. The authors introduce Cluster Directed Acyclic Graphs (C-DAGs), where nodes represent clusters of variables rather than individual variables, simplifying causal discovery in complex systems.

**Questions:**

See the weaknesses above.

**Ethical Concerns:**

["NO or VERY MINOR ethics concerns only"]

**Final Justification:**

The response has addressed my concerns, and I would like to keep the score of acceptance.

**Limitations:**

yes

**Quality:**

3

**Strengths And Weaknesses:**

​​Strengths:​​

- The paper introduces a novel approach to handling causal relationships between variable clusters, improving the scalability for high-dimensional data.
- The authors provide soundness/completeness for d-separation in the cluster of variables in Markovian systems.
- By reducing complexity through clustering, the method could make causal discovery more feasible in domains with many variables.

​​Weaknesses:​​

-  One of my concerns is that while clustering improves scalability, it might introduce statistical challenges. As the number of conditional independence tests grows (with increasing clusters or conditioning sets), the risk of the statistical hypothesis errors​​ escalates, and the conditional independence on the small conditional set should be more trustworthy than the one with have larger one. The paper does not sufficiently address how their method mitigates these errors, which could undermine the reliability of learned structures.
-  Moreover, can you compare the proposed method with the PC algorithm and other causal discovery algorithms without using the cluster by, for example, learn the full causal structure and then inferring the causal graph.
- The approach assumes a predefined partition of variables. Can you provide some real-world examples?

---

> ### Author Rebuttal · Authors · 2025-07-30
>
> Thank you for your thorough evaluation and constructive feedback. We appreciate the recognition of our contributions to causal discovery over clusters and address the specific concerns below.
>
> ***W1:***
> > One of my concerns is that while clustering improves scalability, it might introduce statistical challenges. As the number of conditional independence tests grows (with increasing clusters or conditioning sets), the risk of the statistical hypothesis errors escalates, and the conditional independence on the small conditional set should be more trustworthy than the one with have larger one. The paper does not sufficiently address how their method mitigates these errors, which could undermine the reliability of learned structures.
>
> Causal discovery over clusters, as implemented in CLOC, significantly reduces the number of conditional independence tests compared to variable-level discovery (see Figure 7, Section C.2), leading to greater robustness by minimizing error propagation that can undermine the reliability of the inferred structure. While tests over clusters may have lower statistical power than those over individual variables, this can be effectively mitigated by advances in multivariate testing (e.g., Mantel Test), and modern machine learning-based methods that reliably assess independence between multivariate distributions. Having said that, we appreciate the concern and feel this discussion is indeed very relevant, which will be reflected in the manuscript; thank you for sharing.
>
>
> ***W2:***
>  > Moreover, can you compare the proposed method with the PC algorithm and other causal discovery algorithms without using the cluster by, for example, learn the full causal structure and then inferring the causal graph.
>
> The question asked is unclear to us, so we are happy to elaborate on it further. For now, we would say that there are currently no algorithmic approaches that learn a graphical representation over clusters, as CLOC does; therefore, there is no gold standard against which to compare. A direct comparison between the output of CLOC and that of PC is not meaningful, as they operate over different kinds of entities (clusters vs. variables). If the remark is suggesting to learn the causal structure over variables and then infer the cluster graph, this is what we do with the PC-then-cluster approach, which represents the closest comparator. However, as discussed in Appendix C, the object resulting from PC-then-cluster is a clustered CPDAG, which is distinct from and not the object of interest for CLOC, which is an αC-CPDAG. Alternatively, if the reviewer is asking if our proposed method could be another structural learning algorithm where there are no clusters (where clusters are all of size one variable), we discuss this in Appendix B.3 (our algorithm reduces to PC in this case and therefore works as expected).
>
> ***W3:***
> >The approach assumes a predefined partition of variables. Can you provide some real-world examples?
>
> There are many cases where groupings of variables are natural, and there is, in fact, clarity as to which variables should be clustered together. Medicine represents one domain where this is common; a cluster might be a group of related blood tests in the form of a laboratory panel, related genes, related microbes in a microbiome analysis, anatomical regions in neuroimaging, or demographic information.
>
> We hope the above response clarifies the statistical robustness of CLOC, its comparison to PC, and the realistic applications of the approach.

---

> > ### Comment · Reviewer_GBnh · 2025-08-05
> >
> > Thank you for the response, which has addressed my concerns, and I would like to keep the score.

---

### Decision · Program_Chairs · 2025-09-17

**Decision:**

Accept (poster)

**Comment:**

The paper focuses on an interesting and timely problem in causal discovery, learning (equivalence classes of) causal graphs over clusters of variables, from observational data. The paper is well-written and the theoretical results are rigorous and novel. As mentioned by some of the reviewers, this approach has the potential of helping in scaling up causal discovery by considering clusters of variables instead of individual variables. On the other hand, a limitation of the approach is that the clusters need to be given a priori and it needs to be accurate, which can limit the applicability of the method.

During the discussion, the reviewers have reached a general consensus for recommending an acceptance for this paper based on the above-mentioned strengths and a well-discussed rebuttal by the authors. Overall I agree with their recommendation.